# A subset of viruses thrives following microbial resuscitation during rewetting of a seasonally dry California grassland soil

Alexa M. Nicolas [1], Ella T. Sieradzki [2] ✉, Jennifer Pett-Ridge [3,4], Jillian F. Banfield [2,5,6], Michiko E. Taga [1], Mary K. Firestone [2,6] & Steven J. Blazewicz [3] ✉

Viruses are abundant, ubiquitous members of soil communities that kill microbial cells, but how they respond to perturbation of soil ecosystems is essentially unknown. Here, we investigate lineage-specific virus-host dynamics in grassland soil following "wet-up", when resident microbes are both resuscitated and lysed after a prolonged dry period. Quantitative isotope tracing, time-resolved metagenomics and viromic analyses indicate that dry soil holds a diverse but low biomass reservoir of virions, of which only a subset thrives following wet-up. Viral richness decreases by 50% within 24 h post wet-up, while viral biomass increases four-fold within one week. Though recent hypotheses suggest lysogeny predominates in soil, our evidence indicates that viruses in lytic cycles dominate the response to wet-up. We estimate that viruses drive a measurable and continuous rate of cell lysis, with up to 46% of microbial death driven by viral lysis one week following wet-up. Thus, viruses contribute to turnover of soil microbial biomass and the widely reported $CO_2$ efflux following wet-up of seasonally dry soils.

Soil viruses are abundant and ubiquitous—a gram of soil can hold more than $10^9$ viral-like particles (VLPs)[1,2] and recent studies illustrate their immense taxonomic diversity and potential functions[3,4]. While metagenomic studies hint at many possible functions for soil viruses[5–7], much remains unknown about their quantitative effect on soil microbiome turnover and ecology, especially in response to a perturbation. Viral impact on microbiomes and biogeography is thought to occur through targeted predation shaping microbial community composition and via expression of viral-encoded metabolisms ("auxiliary metabolic genes")[8–11]. In marine systems, viral predation has been approximated to account for killing upwards of ~40% of bacteria daily and redistributing up to 55% of bacterial carbon via the "viral shunt"[9,12,13]. However, in soils, viral-focused metagenomic methods (viromics) is a recent development[14,15] and viral contributions to soil

organic matter turnover or microbial mortality have yet to be quantified[16].

Arid and semi-arid soils cover over 40% of the Earth's terrestrial surface and support more than one-third of the world's population[17]. Many of these systems experience prolonged periods of water deficit, after which, the first rainfall, "wet-up", serves as a pivotal moment which resuscitates soil microbial activity and causes a rapid efflux of mineralized $CO_2$ (the "Birch Effect"[18]) that contributes disproportionately to annual carbon turnover[19,20]. This $CO_2$ pulse has been linked to the rapid growth and death of soil microorganisms[19,21,22], but the mechanisms driving these disturbance-induced successional patterns are unclear. Given observed differential mortality for microbial phyla after rewetting, where *Actinobacteria* and *Proteobacteria* exhibited higher mortality rates than other lineages[19,23], we

[1]Plant & Microbial Biology Department, University of California Berkeley, Berkeley, CA, USA. [2]Environmental Science, Policy & Management Department, University of California Berkeley, Berkeley, CA, USA. [3]Physical and Life Sciences Directorate, Lawrence Livermore National Laboratory, Livermore, CA, USA. [4]Life & Environmental Sciences Department, University of California Merced, Merced, CA, USA. [5]Earth and Planetary Sciences, University of California Berkeley, Berkeley, CA, USA. [6]Lawrence Berkeley National Laboratory, Berkeley, CA, USA. ✉e-mail: ella.shir@gmail.com; blazewicz1@llnl.gov

hypothesized that soil viruses may be induced by rewetting and act both as a top-down control on microbial community assembly and a proximal driver behind the huge wet-up release of mineralized carbon.

The ecological triggers of lysogenic to lytic transitions are not well understood for temperate soil viruses[24,25], but may be stimulated by stress events or shifts in environmental conditions such as wet-up. Lysogenic viral populations can be maintained in soil bacterial genomes as prophages[24,26]. In culture, these prophages often encode an integrase gene that enables phage recombination into the host chromosome[27,28]; upon certain cues, they may excise from host chromosomes and enter a lytic cycle, ultimately lysing their host cell. Induction of prophages can occur due to population stochasticity, external stresses that cause DNA damage, or through quorum sensing pathways[29,30]. In certain marine systems, temperate phages dominate and lysogenic to lytic transitions have been linked to increased bacterial abundance[31], with more integrase-encoding phages found in locations with higher environmental stresses[32]. Given the unique attributes of the soil environment—extraordinary spatial heterogeneity, lack of mixing, microniches of low nutrient availability[33]— some hypothesize that lysogeny may be a predominant lifestyle of soil viruses[2,6,26,30,34,35]. Previous studies have shown that soil moisture modulates soil viral counts, composition, and activity[36,37], and in arid and Mediterranean systems, soil microbes have substantially reduced activity and replication during the summer dry-down[19,21,38,39], possibly favoring lysogeny as a mechanism to persist. Thus, the burst of microbial growth that occurs after soil rewetting may suggest wet-up too serves as an environmental trigger of viruses that had persisted through the summer dry-down as prophages.

Here, we combined time-resolved viromics and quantitative stable isotope probing (qSIP)-informed metagenomics[40,41] to follow dynamics of extracellular viruses (measured as VLPs), and viruses actively infecting host cells. Multiple sequencing approaches (Supplementary Table 1) provided distinct insights into viruses in soil. Metagenomes reveal viruses in multiple life stages: integrated into host chromosomes, viruses in host cells, or extracellular virions. Quantitative stable isotope probing (qSIP) shows the continuum of activity of viruses found in metagenomes and their predicted bacterial hosts. Viromes offer a focused view of extracellular VLPs. Together these techniques enable an understanding of viral dynamics from a total community perspective (metagenomes), from a virion perspective (viromes), and from quantified activity estimates (qSIP).

In this work, we use these data to understand lineage-specific viral-host temporal patterns following rewetting, and how viruses restructure the soil microbiome and contribute to cell death and carbon mineralization. In arid soils, abundant extracellular DNA and dormant cells can mask active microbial populations[42–44], since up to 75% of cells may be inactive at any given time in soil[45]. However, qSIP allows us to directly identify active microbes by tracing a stable isotope ([18]O via $H_2^{18}O$) into the DNA of microbes and viruses actively replicating in response to rewetting[23,46,47]. Measuring the subsequent shift in DNA density[48] enables calculation of the growth and mortality rates of host microbes, and quantification of viral contribution to microbial turnover. We also address the question of whether rewetting serves as a major inducer of prophages. This work builds on our previous efforts to link taxon-specific population dynamics with rainfall-induced carbon fluxes[21,23].

## Results

### Viral succession

We established the set of metagenome-assembled genomes (MAGs) and viral operational taxonomic units (vOTUs) to define the soil microbiome following wet-up from 234 metagenomes over five time points (0h, 24, 48, 72, and 168 h) and 18 triplicated viromes over six time points (0, 3, 24, 48, 72, and 168 h) generated from California grassland soil collected at the end of the dry season (see "Methods" and our paired study[49]). We resolved 377 MAGs (≥50% complete, ≤10% contamination) which we dereplicated to 338 MAGs[49] and combined with 168 dereplicated MAGs from a previous study at our field site, which sampled soil during the previous winter[40]. In total, we studied a combined set of 542 dereplicated MAGs: 503 from total soil metagenomes comprised of diverse soil bacteria and archaea; and 39 from small size-fraction metagenomes[50], i.e., viral-enriched metagenomes or viromes which include genomes from ultrasmall cells such as those of candidate phyla radiation (CPR) bacteria. The total set of 26,368 dereplicated vOTUs in this study represents 229 vOTUs resolved from metagenomes and 26,139 assembled in the virome (Fig. 1). We found that vOTUs in metagenomes and in viromes showed a similar minimum detected relative abundance on the order of the $10^{-8}$ (viromes: $8.2 \times 10^{-8}$, metagenomes: $4.79 \times 10^{-8}$). We consider vOTUs detected in viromes to be extracellular VLPs, which are distinct from vOTUs detected in metagenomes where analysis cannot differentiate between intracellular and extracellular viral populations (Supplementary Table 1). On average, 35.3% (±7.5% standard deviation) of virome reads, and 0.55% (±0.05% standard deviation) of metagenome reads mapped to the total vOTU set. Prior to rewetting, our microbial community was dominated by *Actinobacteria* and *Proteobacteria*[49]. Similar to previous work[20,23], *Proteobacteria* were highly active throughout wet-up and responded early to rewetting, and *Actinobacteria* were also an

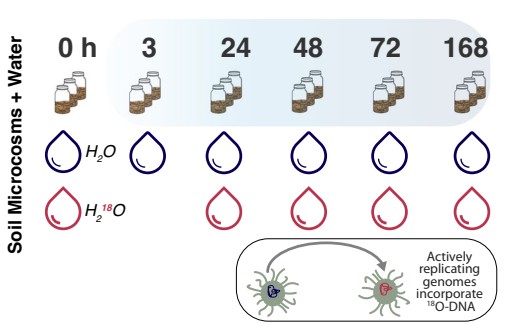

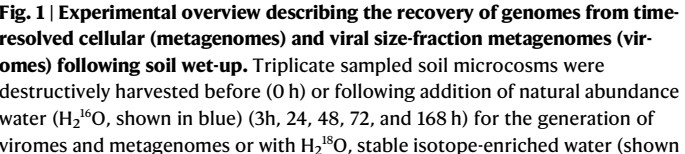

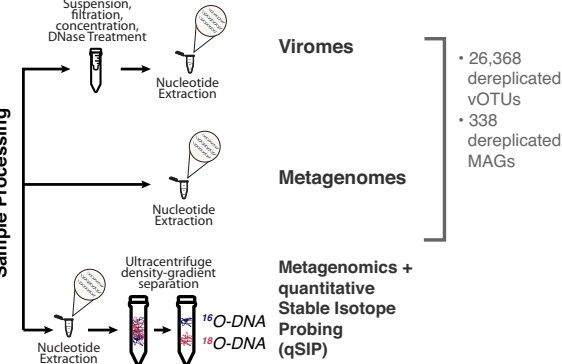

**Fig. 1 | Experimental overview describing the recovery of genomes from time-resolved cellular (metagenomes) and viral size-fraction metagenomes (viromes) following soil wet-up.** Triplicate sampled soil microcosms were destructively harvested before (0 h) or following addition of natural abundance water ($H_2^{16}O$, shown in blue) (3h, 24, 48, 72, and 168 h) for the generation of viromes and metagenomes or with $H_2^{18}O$, stable isotope-enriched water (shown in red), for metagenomics with quantitative stable isotope probing (qSIP). Below the depiction of the microcosms is a cartoon of a microbial cell and its genome before (blue) and after (red) incorporation of [18]O-DNA. Microcosms fated for metagenomic sequencing did not include a 3 h time point. Sample prep for viromes, metagenomes, and qSIP metagenomics prior to shotgun sequencing is shown on the right.

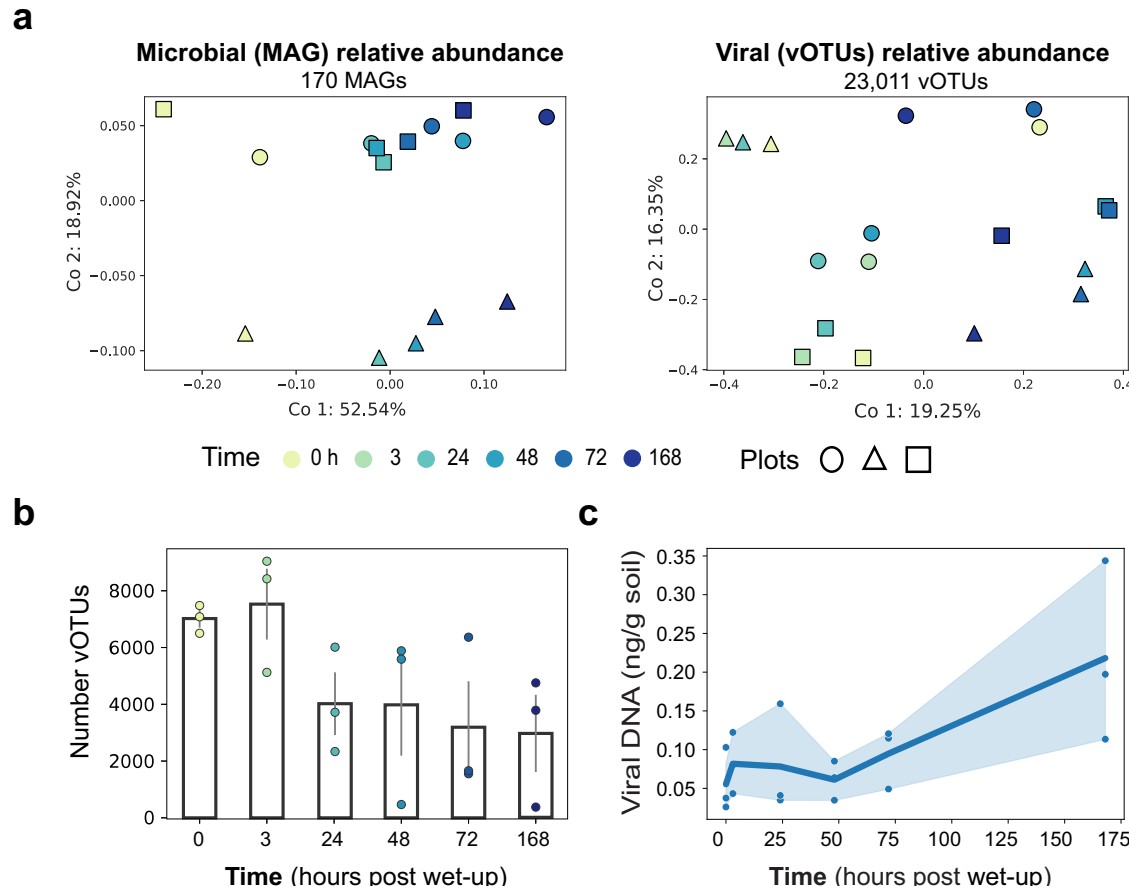

**Fig. 2 | Temporal compositional changes in microbial and viral communities following soil wet-up. a** PCoA plots of MAGs (left) and vOTUs (right) calculated based on Bray-Curtis dissimilarity colored by time with different marker shapes signifying distinct field plots (biological replicates). **b** Barplot of the mean total number of unique vOTUs per time point. Error bars (gray) show the standard error of mean across all three field plots, with underlying points colored by time. **c** Mean viral biomass approximated through time using total extracted virome DNA (ng) per gram of soil normalized by the total vOTU relative abundance. Points represent the calculated viral biomass per distinct microcosm. The shaded area around the line represents a 95% confidence interval.

abundant lineage responding to wet-up, but showed less overall activity despite their higher abundance (see ref. 49 for more details on the microbial and functional response to rewetting).

To visualize the spatial and temporal succession of microbial and viral populations, we calculated ordinations based on the Bray-Curtis dissimilarity of MAG (Fig. 2a, left) and vOTU (Fig. 2a, right) relative abundances. Of our total dereplicated MAGs, 170 met the breadth criteria to be considered present during the first week following rewetting. For MAGs throughout the week following rewetting, the first two principal components revealed that the succession of potential hosts was most pronounced over time (52.54% variance explained) with a minor spatial component and explained in total over 70% of the variability. On the contrary, abundances of viruses had a significant spatial component. The first two principal components of vOTUs represented less variance compared to potential hosts (35.6%). Underscoring the differences in viruses and their host dynamics during this weeklong period, only 25 vOTUs (0.1% of vOTUs in Fig. 2a) appeared in all replicates and time points, whereas 75 MAGs (44.1% of Fig. 2a MAGs) appeared in all time points and replicates (Supplementary Fig. 1A).

Following rewetting, we found that viral richness, as measured through the number of unique vOTUs per time point, significantly decreased, while viral biomass significantly increased. Early time points showed higher richness of vOTUs, but with lower abundances per vOTU (Fig. 2b and Supplementary Fig. 1B). In contrast, later time points, in particular 168 h, shifted to a less even viral community, with

higher relative abundances of fewer vOTUs. Average vOTU richness decreased significantly after 3 h (linear regression−P value = 0.02; $R^2 = 0.28$) (Fig. 2b). While richness decreased by nearly 50% one week post wetting, viral biomass increased by at least fourfold (Fig. 2c). We found viral biomass, as proxied by measuring total extracted viral DNA through time, significantly increased after a week (linear regression−P value = $2.1 \times 10^{-3}$; $R^2 = 0.46$).

To dissect temporal viral succession trends, we focused on the set of vOTUs that were present in at least three time points within each replicate, i.e., "persistent vOTUs." In previous studies, we observed phylum-specific succession of microorganisms in response to wet-up[20,23,49]. We hypothesized that persistent vOTUs would reflect host dynamics and show similar temporal response patterns. We established groups of vOTUs that behaved similarly across wet-up by hierarchical clustering of relative abundances (Supplementary Fig. 3). This clustering demonstrated several broad patterns of abundance dynamics consistent across replicates: (1) vOTUs present early in wet-up (0 h, 3 h, 24 h) that then disappear or drastically decrease in abundance i.e., early vOTUs; (2) vOTUs that appear for the first time at 48 h, i.e., late responding vOTUs; (3) vOTUs that are present across all time points throughout the wet-up, i.e., ubiquitous vOTUs; 4) vOTUs that are present in dry soil (time 0) and at the end of the wet-up time course (168 h), but are not detected in all time points in between, i.e., 0 and 168 h vOTUs. vOTUs that did not fit into these categories were included in the "other" category (Supplementary Fig. 4A). Each of these defined response categories represented up to 15% of the total virome

reads. Overall, we observed that viral successional patterns responding to wet-up followed field plot-specific trends (Fig. 2 and Supplementary Figs. 1, 3, and 4).

We hypothesized that vOTUs would reflect their host responses to rewetting and target *Actinobacteria* and *Proteobacteria* as growing bacteria are expected to be targeted by viruses based on the "kill the winner" hypothesis[51]. While host phylum-level analyses of viruses do not reflect known viral-host ranges (which are thought to be often on a strain level)[1,52], wet-up shows consistent phylum-level microbial responses, and host analyses were limited to coarser taxonomic predictions (i.e., phylum) (see "Discussion"). We examined the vOTU putative host composition within each response group (Supplementary Fig. 4B) and found vOTUs infecting *Actinobacteria* dominated the groups responding immediately and after 24 h. *Alphaproteobacteria*-infecting vOTUs were the second most numerous vOTUs across nearly all response groups. Overall, we found that vOTUs of prevalent hosts in our system (*Actinobacteria* and *Proteobacteria*) appeared as the most prevalent vOTUs in each response category. However, host-specific successional dynamics previously detected of bacteria following wet-up[20,23] were not reflected in our viral-host predictions.

### qSIP informed active virus-host dynamics

To assess the ecological role of viruses during this highly dynamic moment in the soil microbiome, we focused on the active subset of viruses and their microbial hosts. We defined activity as incorporating $^{18}O$ from the $H_2^{18}O$ wet-up (Fig. 1) and applied qSIP formulas to quantitate genome isotopic enrichment (atom percent excess, APE) of DNA[40,41]. We hypothesized that relating viral and host activity following rewetting may reveal to what extent soil viruses follow canonical Lotka–Volterra oscillations, i.e., host activity peaks and precedes peak viral activity[51] (Fig. 3a).

Given the prevalence of MAGs and vOTUs with a taxonomy or host-level taxonomy assignment of *Actinobacteria* and *Proteobacteria*,

and the previous high degrees of mortality and growth following wet-up calculated for these lineages[23], we highlighted analyses comparing vOTUs and MAGs of these lineages. Specifically, within *Proteobacteria*, we focused on *Alpha-* and *Gammaproteobacteria* as growth could not be detected for other classes within this phylum in our experiment. Rather than reflecting Lotka–Volterra predator-prey type dynamics, we observed distinct lineage-specific dynamics (Fig. 3). Active vOTUs predicted to infect *Actinobacteria* showed higher activity than their hosts in the first 48 h, then their activity decreased while host activity increased. This appeared to display a reversed Lotka–Volterra dynamic[53]. Meanwhile, the activity of vOTUs predicted to infect the *Proteobacteria* lineages followed the same temporal trend as host activity did, conceptually shown in the bottom panel of Fig. 3a[54].

The number of detected active MAGs and vOTUs of the same lineages followed similar temporal trends for all three lineages focused on (Fig. 3c). Active vOTUs outnumbered their active putative hosts at nearly all time points. At 24 h, for instance, there were more than twice as many vOTUs as MAGs for all lineages, and for *Actinobacteria* in particular, over fifteen times as many vOTUs were detected as isotopically enriched compared to MAGs. Overall, for both MAGs and vOTUs, we found the greatest number of newly detected active genomes at 48 h. We similarly analyzed other phyla using these metrics of MAG APE, vOTU APE, and number of enriched genomes through time and found no discernable patterns. Of the 177 total enriched vOTUs detected in metagenomes, 58 were detected in viromes, indicating that at least 33% of viruses replicating and assimilating isotope within host cells were also detected as putative virions within the experimental timeframe.

### Viral lifestyle characterization

To discern the mechanism for how viruses may exist, persist, and control host populations following wet-up we investigated the lysogenic potential of vOTUs. To test our hypothesis that wet-up induces

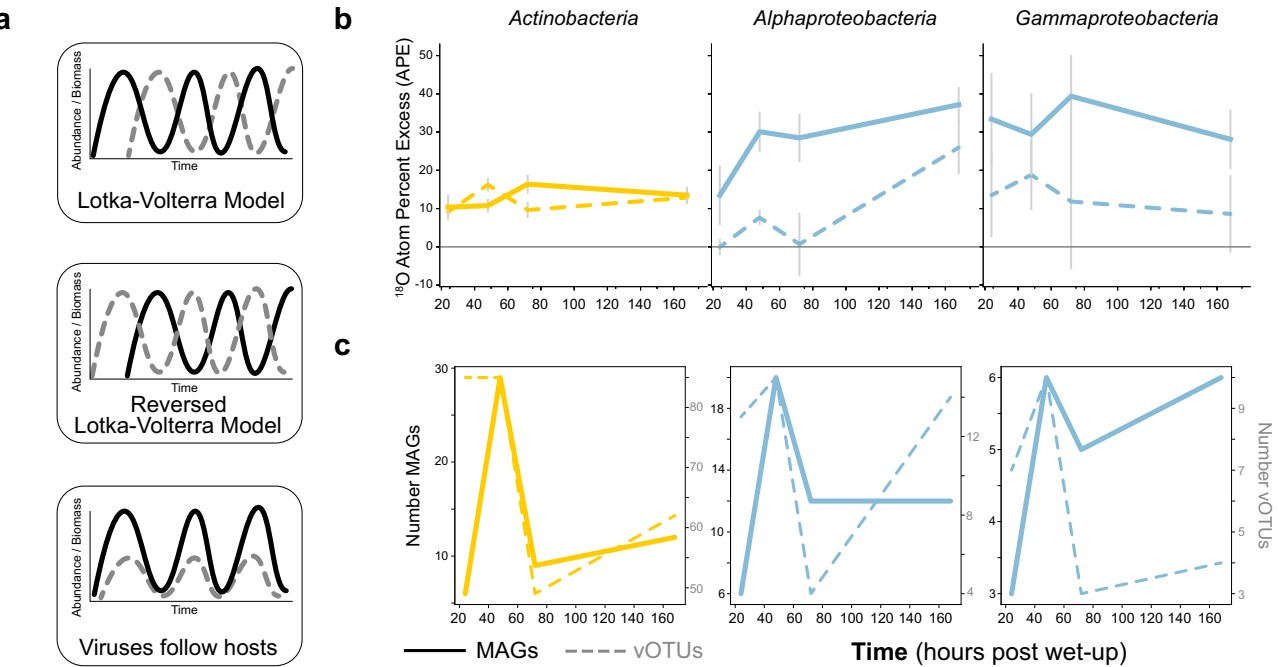

**Fig. 3 | Dynamics of isotope-enriched (growing) viruses and microbes through time following soil wet-up.** Viruses or vOTUs are represented in all plots by dotted lines, MAGs are depicted by solid lines. Colors represent taxonomic groups (yellow are *Actinobacteria*, blue are *Proteobacteria*). **a** Hypothetical models of viral predation on microbes: Lotka–Volterra model, reversed Lotka–Volterra, and a model in which viruses track changes in host abundance. **b** Mean $^{18}O$ atom percent excess (APE) for MAGs and vOTUs through time post wet-up. Error bars represent standard error of genome APE means. **c** Number of unique MAGs and vOTUs per lineage detected as incorporating isotope at each time point post wet-up.

temperate viruses, we combined our sequencing methods to take a: (1) community sequence-centered approach: cross-mapping of vOTUs to metagenome sequences to look for vOTUs in the context of bacterial sequence; (2) vOTU-centered approach: detection of capacity to integrate into a host genome via integrase genes; (3) MAG-centered: prophage detection in MAGs.

First, we searched for vOTU genomes found in both viromes and metagenomes that contained bacterial flanking regions in metagenomes to discover and quantify lysogeny in our system (see "Methods"). This strategy yielded just two candidates of vOTUs found both as VLPs in the virome and as putative prophage in metagenome contigs, and only one of these candidates was successfully detected in a MAG, a *Gammaproteobacteria* of the genus *Pantoea*. The unbinned sequence was predicted to be an *Alphaproteobacteria*, *Rhizobium leguminosarum*. We did not further characterize this binned putative prophage as it was only detected as a VLP in one plot at two non-consecutive time points (0 and 24 h).

Unsuccessful in finding vOTUs appearing as both integrated in MAGs or bacterial sequences and as VLPs, we examined our system for integrating temperate viruses by analyzing the genomic capacity of vOTUs to integrate, via an integrase gene, into a host chromosome (Fig. 4)[55]. Our vOTU set is primarily comprised of vOTUs resolved from virome assemblies, which represent detected extracellular viruses (in a lytic cycle). This approach provided an ideal pool of viruses to test our hypothesis that wet-up serves to induce temperate viruses. Of the total vOTUs detected, 17% (4463) contained an integrase gene and were denoted as putative temperate viruses. Non-integrase-encoding vOTUs represented the majority of total virome sequencing and appeared responsive to wet-up, whereas integrase-encoding vOTUs decreased in proportion following wet-up (Fig. 4a). Further, non-integrase-encoding vOTUs were more numerous at all time points (Fig. 4b).

Concerned that the low detection of integrase genes was due to fragmented assembly of vOTU contigs, we also split the results of

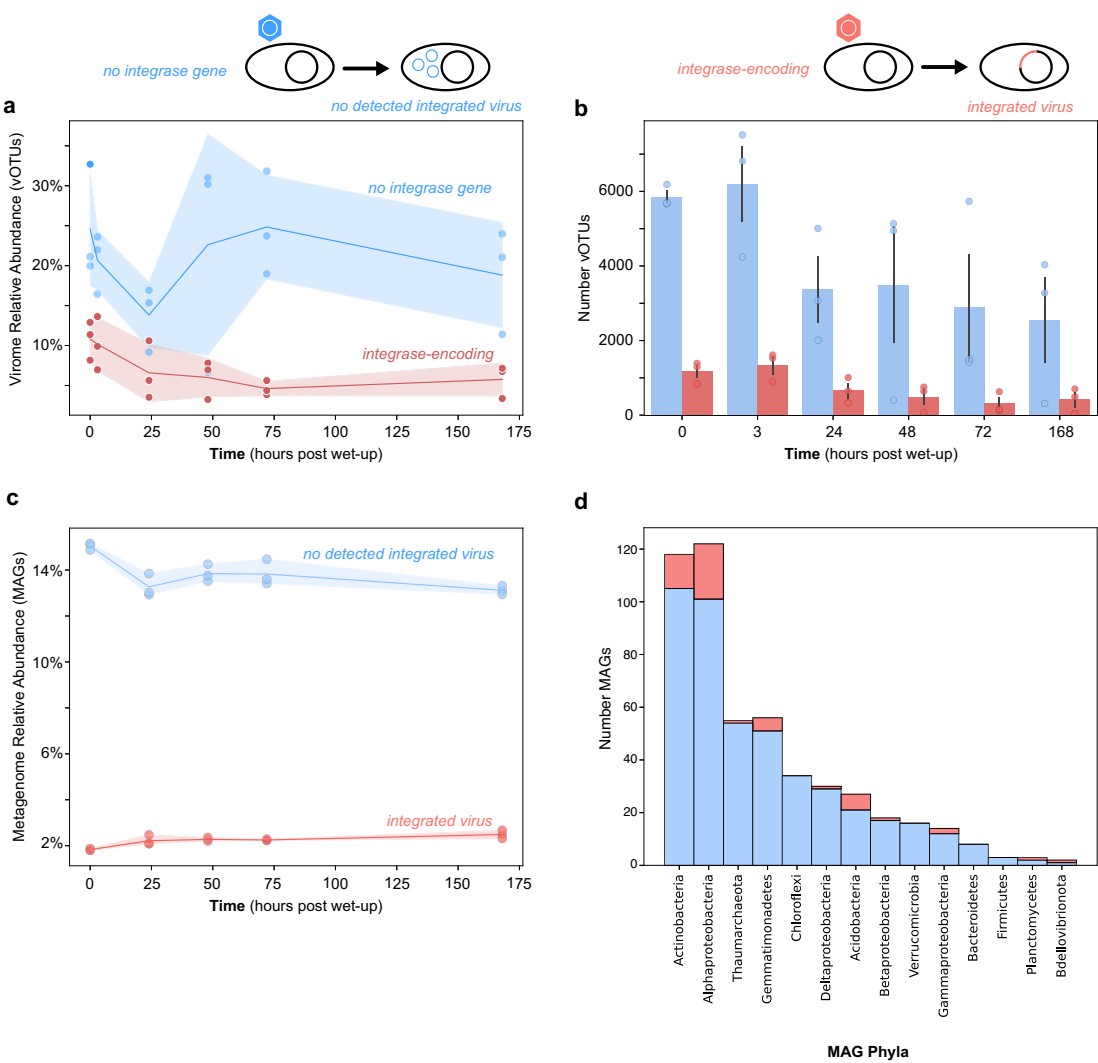

**Fig. 4 | Relative abundance and richness of integrase-encoding vOTUs through time following soil wet-up.** vOTUs were split by whether they encode an integrase gene (red) or not (blue). Cartoon above illustrates integrase-encoding vOTUs (red hexagon) have the capacity for their genomes to integrate into their host's chromosome or no integrase gene vOTUs (blue hexagon) may not be able to integrate their genome into a host's genome. The cartoon depicts vOTUs as hexagons and shows, from a MAG-perspective, detection of an integrated virus or prophage in a host genome. **a** Relative abundance (percent of total sampled reads) of integrase-encoding or not integrase-encoding vOTUs. Shaded area around lines represents a 95% confidence interval. **b** Counts of integrase-containing and non-integrase-containing vOTUs. Error bars show the standard error of the mean number of vOTUs in each of three replicate microcosms. **c** The aggregated relative abundance through time of MAGs detected as containing an integrated virus (red) or not (blue). The shaded area around lines represents a 95% confidence interval. **d** The total number of MAGs per phylum shown in stacked bars the counts containing an integrated virus or prophage (red) and MAGs with no detected integrating virus (blue).

integrase-encoding and non-integrase-encoding viral genomes by vOTUs that circularize, implying genomic completeness, and vOTUs that are linear, and thus may or may not be complete. We found that the linear set of vOTUs followed similar trends to circularized vOTUs, therefore our observed dynamics suggest biological trends rather than artifacts of incomplete assembly (Supplementary Fig. 5).

We hypothesized that if lysogeny was a predominant mechanism of viruses responding to wet-up, then vOTUs appearing after wet-up (0 h) would be enriched in integrase genes—as vOTUs appearing for the first time following wet-up may be prophages excised from genomes. However, post-wet-up detected vOTUs were not significantly enriched in integrase genes compared to vOTUs in dry soil (Chi-square test—$statistic = 1.6$; $P$ value $= 0.2$).

To further support the inference that wet-up does not induce integrating temperate viruses in this system, we took a MAG-centric approach, and predicted prophages in the context of bacterial genomes to discern how many (counts), and how prevalent (abundant) prophages or integrated viruses are in this system (Fig. 4). We analyzed detected prophage-containing MAGs temporally and per phylum using counts of total MAGs containing prophages. Most phyla contained a predicted prophage, but prophage-containing MAGs for each phylum were fewer than MAGs without prophages. MAGs containing integrated viruses were about seven-fold less abundant than MAGs without a detected prophage. We next identified just three prophages that seemed to be actively replicating based on coverage differences between prophage sequences and their flanking host genomic regions[56]. One of these prophages appeared active at all time points in at least two out of the three replicates and was integrated into a MAG of *Alphaproteobacteria* genus *Microvirga*. The other two prophages were active only in a single replicate and in either one or two time points and were integrated into genomes of *Actinobacteria* family *Gaiellaceae* and *Alphaproteobacteria* family *Xanthobacteraceae*, respectively.

To further investigate viral lifestyle and assess whether VLPs correspond to lysis we looked for evidence for *Inoviridae*, known to cause chronic infections that do not lyse the host. We performed an HMM search using the pI-like ATPase protein family HMM, which is conserved in filamentous inoviruses[57], and found no evidence of *Inoviridae* in our viromes or metagenomes.

### Quantifying viral contribution to microbial mortality

We approximated the viral contribution to mortality following wet-up using VLP sequence data from this work and bacterial mortality estimates from corresponding SIP-density fractionated amplicon sequencing[49]. Our calculation integrated 16S rRNA gene-targeted qSIP-estimated microbial mortality rates (most representative of the soil community profile) and virome viral abundances (assaying extracellular viruses to best proxy cell lysis events). We modeled a range of viral burst sizes based on existing literature, ranging from one virus released per cell lysed[2,58] to 200 viruses released per cell lysed[59]. Taken cumulatively, by 168 h viral contribution to microbial death ranged from 0.25% (burst size of 200) to 46.6% (burst size of 1) (Fig. 5b). At 24 h, we observed the highest per day viral contribution to microbial mortality 0.02% (burst size of 200) to 17.4% (burst size of 1).

### Discussion

Soil is a highly complex microbial habitat and home to tens of millions of microbial populations[60,61], where biotic and abiotic factors shape microbial community structure[62–64]. Moments of acute pressure on a community, such as rapid environmental changes or fluxes in resource availability, drive successional trajectories that result in altered community structures; the first rainfall following the dry season, wet-up, is such a moment[19,23,38,65,66]. However, the ecology, lifestyle, and impact of viruses in the soil microbiome during dynamic moments remains unknown. To address this gap in knowledge, we applied viromics and

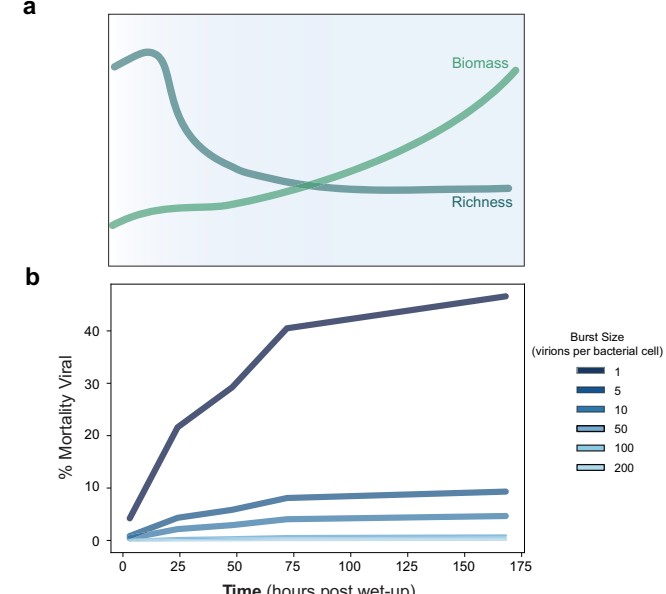

**Fig. 5 | Viral response to wet-up and corresponding contribution to microbial mortality. a** Conceptual model of changes in viral richness (gray-blue) and biomass (green) over time. **b** Modeled contribution of viruses to microbial mortality where each line graphed represents an estimated cumulative percent contribution for a different viral burst size shown in the legend ranging from 1 virion per lysed microbial cell (darkest blue) to 200 virions per lysed microbial cell (lightest blue).

SIP-targeted metagenomics to: (1) assess viral biomass and richness following rewetting and posited that dry soil may serve as a seedbank of virions for the growing season; (2) analyze dynamics of active viruses and found lineage-specific host trends where viruses may follow host populations or control them; (3) investigate viral life cycle and observed minimal evidence for lysogeny suggesting a lytic lifestyle is predominant following wet-up; (4) estimate viral contribution to microbial mortality discovering viral-mediated cell lysis steadily increases following rewetting. We unified metagenomics and viromics to generate a library of vOTUs (Figs. 2–4). We combined these methods to characterize lysogeny in our system as prophages in MAGs (metagenomes), as integrase-encoding vOTUs (metagenomes and viromes), and through cross-mapping of viromes and metagenomes (Fig. 4). We integrated results from 16S rRNA gene qSIP (active bacteria) and virome DNA (estimate of viral particles) to quantify viral contribution to bacterial turnover (Fig. 5). We applied multiple approaches to minimize inherent biases in each sequencing strategy.

In response to rewetting, we observed a large decline in viral richness, and by extension diversity, in the first 24 h, which was inversely related to an increase in viral biomass (Fig. 5a). This pattern was consistent across all soil replicates and thus detectable despite the high degree of inter-sample heterogeneity (Fig. 2a and Supplementary Fig. 1). We observed viral populations may degrade or thrive following rewetting, but after 1-week post perturbation viral composition across field plots (biological replicates) did not become more similar. Rather, the virions present at the end of the dry season determined the trajectory of the plot-specific viral composition. At this timescale and sampling, wet-up is not a perturbation that led to a convergence of viral populations across space despite relative host uniformity at the same sampling scale (Fig. 2). These results imply that dry soil may be a reservoir, or seedbank, of diverse viruses, a subset of which becomes dynamic following rewetting (Fig. 2 and Supplementary Fig. 1B). Following the reduction in viral diversity observed during the first week after rewetting, we hypothesize that the microbial succession that occurs throughout the growing season[67] leads to an increase in soil

viral richness and that these viruses persist as virions during the prolonged dry season when microbial activity is at a minimum. A previous study similarly observed that dryer soils contained more viral clusters[68].

Intriguingly, analysis of the total viral pool responding to wet-up revealed patterns of succession, but viral succession did not appear to be correspondent with known host responses previously reported and observed in our corresponding study (Supplementary Figs. 2 and 3)[20,23,49]. However, focusing on the active subset of microbial and viral populations (Fig. 3), we found taxon-specific trends in viral-host dynamics wherein viruses may follow their microbial hosts or, perhaps, control host populations. Studies from marine systems have shown that viruses may follow their hosts' dynamics[54] and that high local diversity of viruses may be drawn from a seedbank of nearby viruses[9]. As suggested in ref. 54, our isotope targeted results indicate that viruses putatively infecting *Alphaproteobacteria* follow their host dynamics, as opposed to viruses directly controlling host activity or abundances. In contrast, *Actinobacteria*-infecting viruses appear to show a reversed Lotka–Volterra relationship which is suggestive of coevolution-driven dynamics[53]. In this scenario, viral activity is initially higher than host activity indicating that *Actinobacteria* are highly susceptible to viral infection—which selects for viral-resistant hosts as demonstrated by *Actinobacteria* activity increasing after their viruses. Broadly, the total responding pool of putative virions does not necessarily match the total microbial response to rewetting, therefore it is important to focus on the active subset of vOTUs and MAGs to appreciate host-virus dynamics in this soil system. Our study provides a first glimpse into fine spatiotemporal viral dynamics in soil and enabled us to categorize distinct viral responses to rewetting.

Critical to understanding how viruses respond to wet-up and impact the soil microbiome is an analysis of the mechanism of viral life and survival in soil. An emerging idea in soil viral ecology suggests that lysogens, or temperate viruses, are prevalent in soil microbes, and further, that lysogeny is a favorable lifestyle for viruses both to remain linked to a host and to weather conditions difficult for growth, e.g., dry soil[1,2,5,6,29,35,61,69,70]. We tested this hypothesis looking specifically at wet-up as an environmental perturbation that could induce integrated prophage and found no evidence to support this hypothesis.

Multiple metagenomic methods, individually and comparatively, provided complementary analyses to search for potential integrated viruses and their dynamics following rewetting. Taken together, our data suggest that vOTUs which cannot recombine into a host chromosome through an integrase dominate this system. We found no clear evidence showing wet-up significantly induces integrated prophages. The viruses that do increase in abundance following the first 24 h are predominantly non-integrase encoding. As integrating temperate viruses did not predominate as responding viruses to wet-up, prevalent MAGs did not maintain detectable or active prophages (Fig. 4), and mapping extracellular VLPs (viromes) to metagenomes did not reveal additional integrated viruses, we conclude that neither lysogeny as a lifestyle nor integrating temperate viruses in general dominate during wet-up in this soil system. Here we see the power of using metagenomes and viromes as complementary analyses: both perspectives corroborated integrating temperate viruses neither predominate in this system nor in response to soil rewetting. Previous metagenomic studies at this field site[40,71] have not investigated viral lifestyle, and in soil microbiomes at-large, understanding viral lifestyle is understudied.

Whether all detected VLPs in our system originate from cell lysis events remains unclear. While we did not find evidence of *Inoviridae*—viruses that can replicate and release virions without lysing their hosts through a chronic infection[57]—other non-lytic lifestyles may be possible. Further, our methods focused on temperate viruses that may integrate into their host chromosomes. While neither the capacity to integrate into a host chromosome nor persistence as an integrated prophage is prevalent in our wet-up study, we cannot rule out other

mechanisms of viral maintenance in a host cell when a virus is not lytically programmed, such as those of non-integrating temperate viruses like phage-plasmids[27,72] or pseudolysogeny[73]. Our understanding would benefit from future studies that focus on the prevalence and dynamics of phage-plasmids in microbiomes and soil systems[72]. Supporting the prevalence of non-integrating temperate viruses or some other cell-associated state of soil viruses, the majority (67%) of viral populations detected as active were only found in metagenomes (direct DNA extraction from soil) rather than viromes. For a virus to be detected as enriched it must be infecting host cells to replicate, thus assimilating the $^{18}O$ isotope. Most of these active viruses may not be detected in the viromes (i.e., as putative virions) because of their host-associated state. Importantly, the low fraction of persistent vOTUs (Supplementary Figs. 2 and 3) compared to total vOTUs suggests virome-detected viruses may indeed derive from recent host infections.

Microbial death is critical to the flow of nutrients through biomes[61]. A significant feature of soil wet-up is high rates of microbial mortality[19,21,23]. Our previous work demonstrated rapid bacterial mortality following wet-up, with 25% of bacteria degrading in the first 3 h and continued mortality observed through 72 h[21]. We hypothesized that the initial burst of mortality in the first 3 h was driven by osmotic lysis due to the rapid change in osmotic pressure, while mortality after 3 h was driven by viruses or other biological factors[19]. Others have speculated that viruses play a role in observed mortality following rewetting[74]. We found that viruses play a minor role in mortality in the first 3 h, when osmotic lysis is likely the dominant mechanism of lysis, and the contribution to mortality by viruses increases through time, more than doubling after the first 24 h (Fig. 5b). We estimate that viruses can drive a sizeable portion of microbial mortality in soil—by 168 h viral-mediated mortality may contribute 0.23–46.6% of total microbial death dependent on the estimated burst size. While our evidence suggests viruses play an important role in microbial mortality following wet-up, we infer that this influence is minimized as wet-up led to a rapid decline of viral particles in the first 24 h (Fig. 2b, c). If viral biomass continues to increase following 168 h post wet-up, we posit that the viral contribution to microbial mortality may be higher. Our estimate of viral-driven mortality in the soil microbiome aligns with estimates of viral mortality from marine systems ranging from undetectable to majority of mortality, or the more commonly cited estimate via microscopy of 20–40% of bacteria daily lysed by viruses[13,75,76]. There may also be other circumstances in the soil microbiome where viruses play a larger role in mediating microbial death, perhaps in the rhizosphere, where turnover of cells is faster[71,77]. Further studies are required to improve our calculations by constraining the average genome length of soil viruses, range of viral burst sizes in soil, and how long VLPs persist in soil to understand whether virions represent recent infections or whether these calculations require techniques that capture an active set of viruses. Moreover, viral predation has impacts beyond contribution to mortality broadly. Even low viral turnover rates of key lineages may profoundly impact strain selection and functional potential of soil microbiomes[52].

Our data show spatial heterogeneity is an important feature in understanding the response of viruses to rewetting. While a rapid decline in viral diversity, increase in viral biomass, and successional patterns emerged as trends across the studied soil plots, the majority (59%) of viruses appeared in fewer than three of our total 18 viromes and most do not appear in all biological replicates (Supplementary Fig. 1). The patchiness of viruses across plots precluded tracking subsets of viruses through space and time to find statistically meaningful trends. The heterogeneity of soil viruses suggests that infection may not be as uniform across space as potential hosts are. Perhaps, this is the effect of rewetting: virions persisting in soil may be decoupled from their hosts, and the addition of water may exacerbate the hurdle of host-viral physical interactions as previous research has predicted

increased water potential is associated with lower rates of direct cell–cell contact in soil[78]. On the other hand, wet-up may serve to increase connectivity between soil microsites and relieve dispersal limitations of viruses, promoting host encounters. As research has shown, water may constrain microbial or viral dispersal[79,80] which may drive virus-host encounters if they are proximal, or prevent encounters if they are disconnected. Our observed increase of viral biomass following wet-up suggests water would support physical interactions between viruses and their hosts.

The viral heterogeneity observed throughout our field site[81] also raises questions about our own assumptions in the design of this experiment: that the field plot layout, informed by bacterial biogeography would enable similar trends for viruses. Rather, our fine time-resolved study and concurrent work studying heterogeneity on a field scale at longer time intervals[81] have compelled us to question what scale of sampling for viruses represents biological replicates. Our results suggest that future studies should consider the distinct scales of microbial and viral populations in soil to have a holistic understanding of microbial communities[69].

Given the higher spatial homogeneity in microbial distribution and response to rewetting compared to viral distribution (majority of MAGs appear in all samples, Supplementary Fig. 1), we hypothesize that microbes in soil are widely susceptible to infection by many distinct viral populations. However, the lack of specific host-virus linkages in our system prevented investigation of both viral-host range and host susceptibility to multiple viruses.

Linking individual viruses to their specific host genomes remains a challenge in ecovirology. Using mixed methods and multiple databases we were able to determine at least a domain or phylum-level taxonomy for 93% of vOTUs. We found few CRISPR spacer matches[82] and were only able to directly link vOTUs to specific MAGs with high confidence in a few instances. Thus, current limitations in informatics hindered directly implicating viruses as preying on specific MAGs and track predator-prey dynamics on a strain level. Similarly, our viral lifestyle analyses limited us to the identification of prophage only as insertions in host chromosomes which further constrained our ability to match viruses with their hosts. This precluded analysis and understanding of phage-plasmids which persist as extrachromosomal plasmids[72]. However, our phylum-level viral-host predictions revealed that the most numerous and prevalent viruses in our system infected *Actinobacteria* and *Proteobacteria*, which reflected the microbial community, in that the most common viruses infected the most common hosts—implying that viruses may serve an important regulatory role in soil microbial communities. Further, we were able to observe lineage-specific viral-host dynamics on the phylum level which demonstrated that there is not a single mode of viral impact on hosts. Instead, in a complex system, we found multiple models of virus-host dynamics. Taken together, these observations suggest viruses exhibit lineage-specific impacts on host populations, and thus viruses serve an important role in structuring the soil microbiome.

In this work, we probed the wet-up of seasonally dry soil to understand the role of viruses in microbiome assembly. We found that dry soil is a low biomass yet highly rich seedbank of unique virions, and that resuscitation of the microbiome yields a bloom in biomass of a subset of viruses. The succession of the total pool of viruses is driven by spatial heterogeneity. We observed that, on a phylum level, active viral populations show distinct models of impact on hosts. Further, contrary to the emerging paradigm, our data show that lysogeny is not prevalent in our system, underscoring that the reservoir of soil viruses may rely on other forms of host interactions or on virions rather than as inserts in microbial genomes. Our combined use of viromics, metagenomics, and isotope tracing facilitated estimation of the viral contribution to microbial mortality following rewetting. While our conservative calculations indicate viruses may not be the primary mechanism of observed mortality, viruses appear to play an important and dynamic role in the microbial response following wet-up.

## Methods

### Field sample collection

Topsoil samples (0–15 cm, roughly 0.5 m³) from replicate field plots were collected from the Hopland Research and Extension Center (HREC) in Northern California, which is unceded land of the Shóqowa and Hopland People, on August 28th, 2018 after experiencing mean annual precipitation during the rainy season, see our paired study in ref. [49] for full details on field site characteristics. During the growing season the same field plots were covered by mixed grassland flora with *Avena* sp. (wild oat) as the most abundant plant. With permission from HREC, run by UC Berkeley, soil was collected before the first rainfall event of the season, and average gravimetric soil moisture was 3%. The dry soil was transferred to Lawrence Livermore National Laboratory (LLNL) where soil collected from each field plot was individually homogenized and sieved (2 mm) to remove large rocks and roots.

### Wet-up experiment

Stable isotope probing (SIP) is a method for tracking the incorporation of an isotope into the DNA of replicating organisms via ultra-centrifugation (Beckman-Coultier (Indianapolis, IN, USA) VTi65.2 rotor at 44,100 rpm for 109 h at 20 °C) of the DNA in a density gradient. DNA equilibrates in the gradient according to its GC content and isotopic enrichment. The gradient is then divided into density fractions each of which is analyzed for density (handheld AR200 digital refractometer), amount of DNA using the Quant-iT PicoGreen dsDNA Assay Kit (Thermo Fisher Scientific, Waltham, MA, USA), and community composition (shotgun metagenomics). In total, 5 g sieved soil from each of three field biological replicate plots was weighed into separate 15 mL Nalgene flat-bottom vials. Soil was wetted to 22% average gravimetric soil moisture with either $H_2^{18}O$ (98 atom% $^{18}O$) or $H_2O$ (natural abundance SIP controls) (Fig. 1), after which the vials were sealed inside 500 mL mason jars and incubated at room temperature in the dark. Three incubation treatments were established, each in biological triplicate: (1) $H_2O$ amended (used for DNA extraction, unfractionated DNA shotgun sequencing, and as a fractionated DNA control for qSIP); (2) watered with $H_2^{18}O$ for DNA extraction, density centrifugation, and shotgun sequencing (isotopically-labeled fractionated DNA for qSIP); (3) watered with $H_2O$ for viral-like particle (VLP) suspension, concentration, DNase treatment followed by DNA extraction and shotgun sequencing to generate viromes (unfractionated DNA for viromics). Parallel jars were destructively harvested at 0, 3, 24, 48, 72, and 168 h following the water addition; at each harvest, soil vials were frozen in liquid nitrogen and then stored at −80 °C.

### Viral metagenomes ("viromes")

Soil viromes were generated as described in ref. [50] to assess viruses detected as VLPs i.e., extracellularized. Briefly, two 5-gram soil microcosms from each of three field replicates watered with natural abundance $H_2O$ were combined for 10 g soil virome extractions. In total, 1 mL per g soil of a potassium citrate solution (1% potassium citrate, 1× PBS, 100 mM $MgSO_4$, pH 7) was added, mixed by shaking manually and placed on a horizontal shaker (400 rpm, 15 min). Tubes were vortexed for 3 min, manually homogenized for 30 s, and then centrifuged in an Eppendorf Centrifuge 5804 (Eppendorf, Enfield, CT, USA) for 10 min at $4700 \times g$ in a swinging bucket rotor at 4 °C to pellet soil. After pipetting supernatants off into fresh tubes, potassium citrate buffer was added to soil pellets in the same ratio previously and steps were repeated twice more, adding the resulting supernatant to the initial collecting tube. The supernatant was filtered using a 0.22-µm PES membrane-containing Steriflip (EMD Millipore, Vernon Hills, IL, USA) and stored overnight at 4 °C. An Amicon Ultra-15 Centrifugal Filter Unit (MilliporeSigma, Burlington, VT, USA) with a 100 kDa molecular weight

cutoff was used to concentrate the filtered supernatant. Amicon filters were blocked using 2 mL of 1% BSA (0.2-µm filter-sterilized) to coat the filter and were incubated for 1 h at 4 °C. Filters were centrifuged in a swinging bucket rotor at $1500 \times g$ for 10 min (or until all BSA passed through the filter). Excess BSA was removed from reservoirs and filters were washed with 2 mL 1× PBS and centrifuged to remove PBS. Samples were added to coated and washed Amicon devices and centrifuged 5–10 min at a time until samples were concentrated to ~250 µL. Concentrated samples were removed from filter reservoirs by adding 250 µL of the potassium citrate solution in three steps (amounting to a total concentrate of 1 mL). DNase treatment was carried out as in ref. 83: 1 U/µL DNase I (Roche, Indianapolis, IN, USA), and Dnase I Reaction Buffer (100 mM Tris-HCl, 25 mM MgCl₂, 5 mM CaCl₂, pH 7.6) were added to samples and incubated at room temperature for 2 h. To inactivate DNase, samples were treated with EDTA/EGTA to a final concentration of 100 mM. Following DNase treatment to rid nanoparticulate concentrates of free DNA, we extracted DNA from small cells and particles after an iron flocculation step, as described in ref. 84, combined with a phenol:chloroform strategy based on that of ref. 85. In short, 1 µL of 0.02-µm filter-sterilized Iron Chloride Solution (0.1 g Fe per 1 mL deionized water) was added to 1 mL of sample and centrifuged at $14,000 \times g$ for 20 min. The pellet was resuspended in 20 µL of 0.02 µm filter-sterilized Ascorbic-EDTA buffer (0.2 M EDTA, 0.4 M ascorbic acid, pH 6–7).

To extract DNA from the 0.2-µm size-filtered cell and particulate fraction of soil, 250 µL of Phenol:Chloroform:Isoamyl (pH 8) was added to each sample and vortexed for 1 min. Samples were incubated on ice for 15 min, vortexing occasionally to homogenize samples, and centrifuged at 14,000 g for 5 min. The top layer of the aqueous phase was transferred to a Phase Lock Gel (5 Prime, Gaithersburg, MD, USA) and 250 µL of chloroform was added to the tube. Tubes were centrifuged at $14,000 \times g$ for 5 min, and the supernatant was transferred to a fresh tube. Sodium acetate (25 µL, 3 M, pH 5) and 1.5 µL of glycoblue were added and mixed and 250 µL of isopropanol was mixed thoroughly with samples and incubated at −80 °C for 20 min. Samples were centrifuged at $14,000 \times g$ for 20 min, and the supernatant was discarded. The pellet was washed with 500 µL of freshly prepared 70% ethanol and centrifuged at $14,000 \times g$ for 5 min. The supernatant was removed from each sample, and samples were centrifuged for an additional 1 min at $14,000 \times g$. Remaining ethanol was removed from samples and dried for 5 min. Finally, the pellet was resuspended in 15 µL of Tris-HCl. DNA was quantified from all samples by Qubit dsDNA High Sensitivity Assay Kit (Thermo Fisher Scientific, Waltham, MA, USA).

DNA was sent to the DNA Technologies and Expression Analysis Cores at the University of California Davis Genome Center for library preparation and sequencing. The Swift Accel-NGS 1 S Plus DNA Library Kit (Swift BioSciences, Ann Arbor, MI, USA) was used to prepare viromes for sequencing on an Illumina Novaseq (2 × 150 cycles). Due to the low DNA extraction yields, thirteen rounds of PCR amplification were performed using the Swift unique dual index primers, producing libraries averaging 1 ng/µL. Sequencing was performed to attain an average 40 Gb per sample. DNA extraction yields and sequencing metadata are available in Supplementary Data 1. Illumina adapters and phiX were removed from resulting sequences using BBTools v.39.0 (https://jgi.doe.gov/data-and-tools/bbtools/) and then quality-trimmed using Sickle[86]. Previous research has shown that for fewer than 14 rounds of PCR amplification standard assembly and analysis pipelines can be used.[87] Sequences were assembled with MEGAHIT version 1.2.9[88]; we co-assembled all biological replicates per time point using default parameters. Co-assembly of replicate samples may serve as an additional dereplication step at the assembly stage because a consensus sequence will be assembled fewer times per time point. Given differential coverage across replicates per time point, co-assemblies can generate better assemblies[40]. Open reading frames

(ORF) were predicted for each contig using Prodigal v2.6.3[89] and were annotated using sequence similarity searches with USEARCH v10.0[90] to KEGG[91], UniRef[92], and UniProt[93].

### Metagenome-assembled genomes (MAGs) from viromes

To generate microbial bins from the size-fractionated metagenomes, i.e., viromes, we used multiple binning methods. Contigs were binned with default parameters using MaxBin v2.2.7[94], MetaBAT v2.12.1[95], Concoct v1.1.0[96], and manually based on coverage, GC content, and taxonomy using in house tools on ggkbase (https://ggkbase.berkeley.edu). DAS_Tool v1.1.1[97], using default parameters, was used to predict best bins and bins were further manually curated on ggKbase to remove highly divergent sequences based on coverage, GC, and taxonomy. MAGs from viromes were clustered and dereplicated alongside MAGs from metagenomes (discussed below) using dRep v3.0.1[98], with default parameters, requiring at least 75% completeness and no more than 10% contamination, yielding 39 genomes from candidate phyla radiation bacteria and archaea.

### Viral prediction and generation of viral operational taxonomic unit (vOTU) set

To establish a set of viral genomes from our metagenomic efforts we used multiple methods of prediction and dereplicated vOTUs across samples. Viruses were predicted from metagenome and virome contigs. Viruses were predicted using default parameters of the following programs: VirSorter1 v1.0.6[99] (all categories), VirSorter2 v2.2[100] (dsDNA and ssDNA only), VIBRANT v1.2.1[101], deepvirfinder v1.0[102] (score ≥0.9; $P$ value ≤ 0.05), and seeker v1.0.3[103]. From the set of virome and metagenome contigs explored 3,979,792 were predicted to be viral. To most robustly establish a viral set we chose to subset these predicted viral sequences to only those identified by more than one tool[104], 777,725 contigs met these criteria, and 32,660 of these sequences were either predicted to be circularized contigs by VRCA[105] (i.e., implied to be a complete genome), or ≥10,000 base pairs long. This set of size-filtered, circularized, repeatedly predicted viruses was dereplicated using MMseqs2 v13-45111[106] (95% ANI, 85% breadth required of sequences) to 26,368 viral operational taxonomic units (vOTUs). All vOTUs were annotated using DRAM-v v1.2.0 with default parameters[106]. To assess viral lifestyle we searched DRAM-v output for integrase-annotated ORFs and denoted integrase-containing vOTUs as temperate viruses with the capacity to integrate into a host chromosome. A single-gene approach will not capture all temperate viral diversity, but integrase genes have been used as an effective hallmark with much higher recovery than other genes (e.g., excisionases) in previous peer-reviewed studies[27,34,55,68,107,108]. To identify potential Inoviridae in our vOTUs we performed HMM searches of known pI-like ATPase protein family using 32 protein models constructed based on alignments of pI-like ATPase from Inoviridae identified in RefSeq[109], publicly available (https://github.com/simroux/Inovirus/blob/master/Inovirus_classifier/Ino_classifier_db/pI_PCs_db_annot.hmm). We used HMMER v3.1b2 (http://hmmer.org) with default parameters i.e., $e$-value ≤ 10. No viral proteins met these lax parameters to be considered homologous to Inoviridae pI-like ATPases.

### DNA density fractionation

For downstream qSIP analyses and non-virome metagenomes, DNA was extracted from three biological replicates at each time point using a modified phenol-chloroform protocol adapted from ref. 110. In brief, DNA was extracted in triplicate from soil microcosms and replicate DNA extracts were combined. For each extraction, soil (0.4 g +/− 0.001) was added to 2 mL Lysing Matrix E tube (MP Biomedicals, Santa Ana, CA, USA) and extracted twice as follows: 500 µL extraction buffer (5% CTAB, 0.5 M NaCl, 240 mM K₂HPO₄, pH 8.0) and 500 µL 25:24:1 phenol:chloroform:isoamyl alcohol were added before shaking (FastPrep24, MP Biomedicals: 30 s, 5.5 m s⁻¹). After centrifugation

(16,100 g, 5 min), residual phenol was removed using pre-spun 2 mL Phase Lock Gel tubes (5 Prime, Gaithersburg, MD, USA) with an equal volume of 24:1 chloroform:isoamyl alcohol, mixed and centrifuged ($16,100 \times g$, 2 min). The aqueous phases from both extractions were pooled, combined with 7 μL RNAase (10 mg/ml), mixed by inverting, and incubated at 50 °C for 10 min. 335 μL 7.5 M $NH_4^+$ acetate was added, mixed by inverting, incubated (4 °C, 1 h) and centrifuged ($16,100 \times g$, 15 min). The supernatant was transferred to a new 1.7-mL tube and 1 μL Glycoblue (15 mg/ml) and 1 mL 40% PEG 6000 in 1.6 M NaCl were added, mixed by vortexing, and incubated at room temperature in the dark (2 h). After centrifugation ($16,100 \times g$, 20 min), the pellet was rinsed with 1 mL ice-cold 70% ethanol, air-dried, resuspended in 30 μL 1× TE and stored at −80 °C.

Samples were density fractionated in a cesium chloride density gradient formed by physical density separation in an ultracentrifuge as previously described[23], with minor modifications. For each sample, 5 μg of DNA in 150 μL 1xTE was mixed with 1.00 mL gradient buffer, and 4.60 mL CsCl stock ($1.885 \text{ g mL}^{-1}$) with a final average density of $1.730 \text{ g mL}^{-1}$. Samples were loaded into 5.2-mL ultracentrifuge tubes and spun at 20 °C for 108 h at 176,284 $RCF_{avg}$ in a Beckman Coulter Optima XE-90 ultracentrifuge using a VTi65.2 rotor. Automated sample fractionation was performed using Lawrence Livermore National Laboratory's high-throughput SIP pipeline "HT-SIP"[41], which automates fractionation and clean-up tasks for the density gradient SIP protocol. Ultracentrifuge tube contents were fractionated into 36 fractions (~200 μL each) using an Agilent Technologies 1260 isocratic pump (Santa Clara, CA, USA) delivering water at 0.25 mL min$^{-1}$ through a 25 G needle inserted through the top of the ultracentrifuge tube. Each tube was mounted in a Beckman Coulter fraction recovery system (Brea, CA, USA) with a side port needle inserted through the bottom. The side port needle was routed to an Agilent 1260 Infinity fraction collector. Fractions were collected in 96-well deep-well plates. The density of each fraction was measured using a Reichart AR200 digital refractometer (Depew, NY, USA) fitted with a prism covering to facilitate measurement from 5 μL[111]. DNA in each fraction was purified and concentrated using a Hamilton Microlab Star liquid handling system (Reno, NV, USA) programmed to automate glycogen/PEG precipitations[41]. Washed DNA pellets were suspended in 40 μL of 1xTE and the DNA concentration of each fraction was quantified using a PicoGreen fluorescence assay.

All DNA directly extracted from soil i.e., metagenomes, were sequenced on an Illumina Novaseq platform 2 × 150 cycles at Novogene (Sacramento, CA, USA). Pre-aliquoted unfractionated DNA from each sample was sequenced to an average of 20 Gb per sample, and DNA from each binned group of density fractions was sequenced to an average of 10 Gb per binned group, yielding a total of 227 metagenomes. Metadata is available in Supplementary Data 1. Illumina adapters and phiX were removed using BBTools and reads were quality-trimmed using Sickle using default parameters.

### Metagenome assembly, annotation, and binning
To establish MAGs from metagenomes we approached assembly in two ways: (1) co-assembly of unfractionated triplicates for each time point; (2) co-assembly of the 5 density fractions from each time point and each replicate. We used MEGAHIT version 1.2.9 for assemblies[88] with preset large-meta and disconnect ratio 0.33[49]. MetaWRAP v1.3.2[112] was used to map reads from unfractionated metagenomes to contigs from each assembly, bin each assembly with MetaBAT2 v2.12.1[95] and MaxBin2[94] and refine bins. Genomic bins from all samples as well as bins from soil metagenomes collected at the same site in the preceding winter (https://ggkbase.berkeley.edu/wsip-metawrap-drep-bins/organisms)[40] were then dereplicated at 95% average nucleotide identity (ANI) using dRep v3.0.1[98] to create a final collection of high quality 503 metagenome-assembled genomes (MAGs) that were on average 87% complete and

5% redundant. Taxonomy was assigned to bins using GTDB-tk v1.5.1[113].

To calculate abundance measurements for MAGs, reads from unfractionated samples were mapped back to the MAG collection using BBmap v.39.0 with minid=0.98 (https://jgi.doe.gov/data-and-tools/bbtools/). Mapping statistics were then used to calculate coverage at reads per kilobase million (RPKM), mapped read counts and breadth of coverage using CoverM v0.6.1 (https://github.com/wwood/CoverM), and were normalized to sequencing depth.

### Calculating relative abundances of the vOTU collection
To quantify abundance of vOTUs in viromes and metagenomes and to assess activity via qSIP we mapped reads from these sequencing efforts to our vOTUs. All reads from viromes, metagenomes, and SIP-fractionated metagenomes were mapped to the set of vOTUs using bbwrap.sh at 98% minimum identity. SAMtools v1.17[114] sort and index functions were applied to the resulting bam files. We used CoverM to calculate breadth and read count per vOTU per sample. Read counts were divided by total reads per sample to calculate relative abundance. For virome relative abundance we considered vOTUs present in a sample if they had at least 80% breadth covered (fraction of genome covered by reads). We found that few vOTUs had reads that mapped stringently (98% identity) and were found in all three replicates of multiple density fractions in both $^{18}O$ and natural abundance $H_2O$ treatments (Supplementary Fig. 2). Thus, we relaxed the breadth cutoff to 50% as it appeared to be an inflection point between breadth and total vOTUs mapping which balanced between falsely excluding vOTUs that appeared in triplicate in multiple density fractions at a high mapping ID, and falsely including erroneously detected present vOTUs (Supplementary Fig. 2). vOTUs meeting this lower breadth cutoff, 50%, in triplicate for a given sample were treated as present in SIP fractions and were analyzed for activity with qSIP calculations (described below).

### Ordination plots
To understand compositional differences through time responding to rewetting, we constructed ordination plots. For MAGs we constructed a matrix of metagenome relative abundance per sample per genome. Similarly for vOTUs, we generated a matrix of relative abundance per sample per genome, using vOTU relative abundance as calculated from our virome samples. We used the python scikit package[115] to calculate Bray-Curtis dissimilarity and to perform principal coordinates analysis (PCoA) to visualize the dissimilarity between samples.

### Virus-host linkage and assigning viruses a host taxonomy
To link vOTUs to potential host microbial MAGs, we used multiple approaches, starting with CRISPR spacer searches. We predicted CRISPR arrays using MinCED v0.4.2 (https://github.com/ctSkennerton/minced)[116] in the original set of MAGs (not dereplicated) from this study, in MAGs from another qSIP study conducted in a soil from our field site[40], and in our viromes. 84 MAGs and 32 vOTUs contained CRISPR arrays with at least 1 spacer. We aligned the MAG spacers against the vOTUs using BLASTn v2.12.0[117], and vice versa, and screened for hits with no more than one mismatch and no gaps between the full spacer and the vOTU. By these criteria, only one vOTU spacer matched a MAG in our dataset. This vOTU appeared in two of three plots and in two time points for each plot. In contrast, 25 MAG spacers from 6 MAGs matched 19 vOTUs. From these linkages we assigned host taxonomies to vOTUs using the associated MAGs GTDB taxonomy[113].

To assign host taxonomy to additional vOTUs we combined these CRISPR spacer searches in MAGs, with a broader set of CRISPR spacer matches, and contig taxonomy predictions. In addition to MAG and vOTU spacers, we also predicted spacers in unbinned metagenome contigs using MinCED and established contig taxonomy for these

unbinned contigs using Kaiju v1.8.0[118]. We used the same BLAST parameters to match these unbinned CRISPR array spacers to our dereplicated vOTU set. Next, we queried the comprehensive spacer database generated in ref. 119 against our vOTU sequences using the same BLAST parameters. Each spacer from this database[119] included an NCBI taxonomy that was assigned to matching vOTU protospacers as the host taxonomy.

We also used vOTU sequences to inform host taxonomy. Previously, Al-Shayeb et al. and others assigned host phyla to phages based on whether ORF annotations in phages were predominantly of one phylum[120]. Here, we used Kaiju on vOTUs to assess taxonomy. Most vOTUs were assigned to microbial taxonomies that presumably reflect shared host-virus genome/gene content. We used this microbial taxonomic assignment to set a putative host. For vOTUs predicted to have viral taxonomies, we captured host lineages using the Virus–Host Database (https://www.genome.jp/virushostdb/)[121].

Lastly, we created a database of all vOTUs, their host predictions, and the source of each host prediction—CRISPR spacer-based taxonomy from GTDB for MAGs, unbinned spacers with Kaiju taxonomic predictions, NCBI taxonomies from spacers matched from the spacer database used, or Kaiju taxonomy predictions per vOTU. We used this database to determine the consensus taxonomy between these sources where there were multiple host predictions closest to a species level by searching for the greatest common denominator level of taxonomy. If consensus was not found between host predictions at any level of taxonomy, host taxonomy would be assigned as "unknown." As the majority of vOTUs neither contained matched spacers nor protospacers, most vOTUs contained only a single host prediction from Kaiju, which was retained. This allowed us to find putative hosts for 93.1% of total vOTUs at the domain level, and 92.8% at the phylum level. We also attempted to link specific MAGs and vOTUs using a network-based analysis modeling canonical predator-prey dynamics, but were unsuccessful in ground truthing these linkages and thus did not include these host matches.

### Examination of evidence for viral induction

The following methods largely describe negative results, however we chose to disclose these methods to provide information for method developments for the benefit of the scientific community. To quantify viruses in our system that were present both as prophages in MAGs and as VLPs in the virome we mapped virome reads to metagenomes and looked for alignment between metagenome sequences and vOTUs. Specifically, we mapped all virome reads to metagenome-assembled sequences at 90% average nucleotide identity. We used coverM v0.6.1 (https://github.com/wwood/CoverM) to generate breadth and coverage calculations for mapped reads and identify metagenome contigs that had virome reads mapping with at least 1000 bp covered by reads, but not mapping with 100% breadth. We reasoned this may indicate that reads mapped to a potential lysogenic viral genome embedded in the host genome. Using BLASTn v2.12.0[117], we aligned the set of metagenome contigs that met these criteria against our vOTUs with a minimum e-value of $10^{-100}$. We subsetted the resulting matches to vOTUs that matched a metagenome contig with greater than 90% nucleotide identity, aligned with a metagenome sequence that had more than 10 Kbp of flanking region around the alignment, and an aligned section of at least 10 Kbp. This resulted in 11 vOTUs representing candidate lysogenic viruses. Using ggKbase (https://ggkbase.berkeley.edu/), we manually analyzed each of these vOTUs and their corresponding metagenome contigs for aligned and unaligned regions consistent with phage (enriched for hypothetical genes, viral structural genes, integrase genes). This resulted in two sequences that appeared to be lysogenic: found both as vOTUs in viromes (presumably in a lytic stage) and in metagenomes in the context of bacterial sequences (hypothetically as integrated prophage).

### Assessment of prophages in MAGs

Prophages were identified from our set of MAGs by VirSorter1 v1.0.6 categories 4 and 5[99], or by VIBRANT v1.2.1[101]. Active prophages, i.e., prophages assessed as replicating more than their flanking host genome, were identified by running PropagAtE v1.1.0[56] using default parameters with metagenome read mapping files to MAGs previously generated. PropagAtE relies on differential coverage between prophage and flanking host regions. Identified prophage-containing MAGs were manually matched from their GTDB to a NCBI taxonomy (to align with vOTU host match names). We established the count of MAGs per phylum that contained or did not contain a prophage. We summed the relative abundance of MAGs by whether the MAG had a detected prophage to generate a seaborn lineplot overlaid with a scatterplot of relative abundances split into these categories.

### Viral response categories

To establish viral response patterns through time, we constrained the set of vOTUs to 6,840 vOTUs present in a specific field plot for three or more time points; we refer to these as "persistent vOTUs." We generated hierarchically clustered heatmaps (cluster maps) per field plot in seaborn[122] using relative abundances per time point for each vOTU. With cluster map observations and k-means clustering, we established 5 response categories based on vOTUs presence or absence in given time points: (1) "early vOTUs" present at 0, 3, and 24 h post wet-up; (2) "late responding vOTUs" present at 48, 72, and 168 h post wet-up; (3) "ubiquitous vOTUs" present in all time points; (4) "vOTUs present at 0 h and 168 h" and one other time point; (5) "other" vOTUs that did not fall into these categories. We summed vOTU relative abundances across these categories for each time point per plot. We used the seaborn relplot function[122] to visualize response categories through time per plot. To measure the host signal by response category, we summed the unique vOTUs of a given bacterial putative host phylum or domain per category. We visualized host abundances per response category by overlaying seaborn swarmplots on boxplots where each point corresponds to a field plot.

### Construction of integrase-containing vOTUs relative abundance and richness barplot

We used the seaborn relplot function[122] to construct graphs of vOTUs that did (4463 unique vOTUs) or did not contain integrase genes. We plotted the set of vOTUs split into these two categories and by circular versus linear, by their aggregated relative abundance through time, and by vOTU richness per category through time.

### Quantitative stable isotope probing

Atom percent excess (APE) isotopic enrichment values were calculated using quantitative stable isotope probing (qSIP)[40,46,49] by normalizing MAG and vOTU relative abundance to the quantity of DNA per density fraction[40]. Both bacterial and viral genomes were required to be detected in all three replicates of each labeled $^{18}O$ treatment and in three replicates of unlabeled (natural abundance $H_2O$) treatments for the bootstrapped median atom percent excess (APE) to be calculated per genome. Formulas used to calculate APE can be found on GitHub, https://github.com/bramstone/qsip. From the qSIP output, we removed genomes with an APE median confidence interval lower than 0 (i.e., not statistically significant). We plotted the average median APE MAG and vOTU values through time for our most prevalent vOTU and host phyla—Actinobacteria, Alphaproteobacteria, and Gammaproteobacteria—using seaborn line relplots. We enumerated the number of detected enriched vOTU and MAG genomes and plotted these counts per phylum through time.

## Viral contribution to microbial mortality

We modeled the viral contribution to microbial mortality using a range of potential burst sizes (virions released per infected cell) to relate the number of approximated infectious viral particles through time to observed microbial cell death counts. Counts of microbial cell mortality were derived from our qSIP analysis of corresponding amplicon sequence data[49] as they best represent the community profile. Cumulative bacterial mortality was calculated using qSIP-estimated mortality rates of 16S rRNA gene sequence and qPCR data as described in ref. 23. In brief, 16S rRNA genes were sequenced from 330 SIP-density fractions amplified in triplicate 10 μL reactions using primers 515 F (5′-GTGYCAGCMGCCGCGGTAA) and 806 R (5′-GGACTACNVGGGTWTCTAAT)[123,124]. Each reaction contained 1 μL sample and 9 μL of Phusion Hot Start II High Fidelity master mix (Thermo Fisher Scientific, Waltham, MA, USA) including 1.5 mM additional $MgCl_2$. PCR conditions were 95 °C for 2 min followed by 20 cycles of 95 °C for 30 S, 64.5 °C for 30 S, and 72 °C for 15 S. The triplicate PCR products were then pooled and diluted 10X and used as a template in a subsequent dual indexing reaction that used the same primers including the Illumina flowcell adapter sequences and 8-nucleotide Golay barcodes (15 cycles identical to initial amplification conditions). Resulting amplicons were purified with AMPure XP magnetic beads (Beckman-Coultier, Indianapolis, IN, USA) and quantified using the Quant-iT PicoGreen dsDNA Assay Kit (Thermo Fisher Scientific, Waltham, MA, USA) on a BioTek Synergy HT plate reader (Agilent, Santa Clara, CA, USA). Samples were pooled at equivalent concentrations, purified with the AMPure XP beads, and quantified using the KAPA SYBR FAST qPCR kit (KAPA Biosystems, Cape Town, South Africa). Libraries were sequenced on an Illumina MiSeq instrument at Northern Arizona University's Genetics Core Facility using a 300-cycle v2 reagent kit.

We used the median rate of 16S rRNA copies lost per gram soil per day and converted this rate to per gram soil by multiplying by the number of days. To convert 16S rRNA copies to cells lost, we used a conservative assumption of six rRNA copies per microbial cell[21,125]. The approximate loss of microbial cells per gram of soil was used to represent the total observed microbial mortality. To calculate the contribution of viruses to observed mortality, we estimated the number of total virions per time point, using the mean aggregated relative abundance across plots (total reads mapped to our vOTU dataset). We used the fraction of viral reads to estimate viral DNA of total extracted DNA per time point per plot. With our estimate of viral DNA and a calculated mean genome length in our vOTU dataset of $2.1254 \times 10^4$ bp, we used equation 1 (previously used to estimate plasmid copy numbers from DNA) to calculate virions per time point:

$$\text{Virions} = \frac{X_{ng} \times 6.0221 \times 10^{23} \frac{\text{molecules}}{\text{mole}}}{N \times 660 \frac{\text{g}}{\text{mole}} \times 1 \times 10^9 \frac{\text{ng}}{\text{g}}} \quad (1)$$

$X = Calculated\ viral\ DNA$
$N = Average\ calculated\ genome\ length\ (bp)$

As the average viral burst size in soils is unknown, we converted number of virions to cells lysed by using a range of viral burst sizes from one virus released per cell[2,58] to 200[59]. Virions scaled to cells dead by the burst size range served as a numerator for viral contribution to microbial mortality calculated with Eq. (2). We next captured cumulative viral contribution to microbial mortality through time by taking the cumulative sum of all previous time points for each time point.

$$\text{Viral contribution to microbial mortality} = \frac{\text{Virions} \times \text{burst size}}{\frac{\text{degraded rRNA copies}}{6} \times \text{Days}} \quad (2)$$

Our calculations made the following assumptions: (1) All virions detected lyse their host. (2) Virions calculated for each time point represent new infections rather than virions persisting through time i.e., virions were released from newly lysed cells. (3) Our VLP-virome extraction protocol efficiently extracted 100% of soil viruses, i.e., viruses detected represent all possible soil viruses. (4) The mean vOTU genome length represented an accurate approximation of soil viral lengths irrespective of short-read metagenomics and potential assembly fragmentation. Our data supported the second assumption given that only a minority of vOTUs appeared in multiple time points (Supplementary Fig. 1). Further, we performed a sensitivity analysis of the impact of genome length on our calculation and found a statistically significant relationship between genome length and percent viral contribution to mortality (linear regression on log-transformed data, slope: −1.00, intercept: 7.80, r value: −0.73, P value: $6.63 \times 10^{-07}$)—changing viral genome length by order of magnitude led to an order of magnitude decrease in the percent contribution to mortality.

### Reporting summary

Further information on research design is available in the Nature Portfolio Reporting Summary linked to this article.

## Data availability

DNA extraction data and sequencing metadata data generated in this study are provided in the Supplementary Information/Source Data file. Metagenome and virome sequencing reads generated in this study have been deposited in the NCBI SRA database under accession code PRJNA856348. Viral genome sequences i.e., vOTUs are available from ggKbase [https://ggkbase.berkeley.edu/hopland_4th_wedge_virus_set/]. All processed qSIP data per genome and sequence (16S amplicon, MAGs, and vOTUs), relative abundance data for both vOTUs (in viromes and metagenomes) and MAGs (in metagenomes), and viral-host taxonomy data are available in the associated GitHub repository [https://github.com/amnicolas/soilviralwet-up][126]. Additional viral-host matches were generated using the Viral-Host Database [https://www.genome.jp/virushostdb/] and the database CRISPR spacer database in Shmakov et al., 2017[119] [https://doi.org/10.1128/mBio.01397-17].

## Code availability

Code can be found on GitHub, https://github.com/amnicolas/soilviralwet-up. No custom tools were used to analyze the data.

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

## Acknowledgements

This research was supported by the U.S. Department of Energy (DOE), Office of Biological and Environmental Research (BER), Genomic Science Program Lawrence Livermore National Laboratory (LLNL) "Microbes Persist" Soil Microbiome Scientific Focus Area SCW1632, and a subaward to UC Berkeley. Field plots and precipitation management were initially generated via DOE BER awards DE-SC0020163 and DE-SC0016247 (to M.K.F.) and awards SCW1589, and SCW1421 (to J.P.R.). Work at Lawrence Livermore National Laboratory was conducted under the auspices of the U.S. DOE under contract DE-AC52-07NA27344. We would like to thank Rachel Hestrin and Alexander Jaffe for helpful conversations regarding data analysis and visualization, and interpretations of isotopic data. We thank Katerina Estera-Molina for her expertise and management of the Firestone Lab fieldwork site at the Hopland Research and Extension Center (HREC). Members of the Taga and Firestone labs provided helpful commentary and criticisms of this work. In particular, Mengting Maggie Yuan engaged in important conversations on statistical analyses and Peter Chuckran helped in manuscript proofing. Rohan Sachdeva and Shufei Lei of the Banfield lab equipped this project with the necessary computational power and data management via the biotite server and ggKbase, respectively. The authors would also like to thank Simon Roux for assistance with the identification of Inoviruses. Support staff at LLNL enabled the large-scale wet-up microcosm experiment and sample extractions, particularly Xiao Bin Max Li and Marissa Lafler. We acknowledge that HREC sits on the traditional, unceded land of the Pomo Indians. Research conducted at the University of California, Berkeley took place on the territory of xučyun (Huichin), the ancestral and unceded land of the Chochenyo-speaking Ohlone people, the successors of the sovereign Verona Band of Alameda County. This land was and continues to be of great importance to the Muwekma Ohlone Tribe and other familial descendants of the Verona Band.

## Author contributions

S.J.B., M.K.F., J.P.R., and A.M.N. designed the large-scale wet-up experiment; S.J.B. led the wet-up experiment supported by A.M.N. and LLNL support staff. A.M.N. directed the generation of viromes from wet-up soil microcosms and analyzed subsequent sequence data to identify and establish a set of vOTUs from viromes and metagenomes and performed analyses described in the paper. S.J.B. directed sample processing on the HT-SIP pipeline and generation of metagenomes. E.T.S. generated applied qSIP calculations for the set of MAGs, identified integrase annotations in vOTUs, and prophages in MAGs. J.F.B. provided the informatics pipelines and computational resources on the biotite server. A.M.N. wrote the manuscript and created all figures with guidance from S.J.B., J.P.R., M.E.T., M.K.F., and J.F.B. All authors contributed to manuscript revisions and gave final approval for publication.

## Competing interests

J.F.B. is a founder of Metagenomi. The remaining authors declare no competing interests.
