## [Peer Review File · Nature Communications]

nature portfolio

Peer Review FilePeer Review comments, first round -

Reviewer #1 (Remarks to the Author):

General Comments

This manuscript by Nicolas et al. describes work to characterize the response of bacteria and their associated viruses to wet-up, a phenomenon that occurs when water is added to dry soils. The response of dryland soil bacteria to wet-up has been investigated for decades, with increasing sophistication as methods expand. Very few studies have characterized the response of soil viral populations, nor their interactions with bacterial hosts during this process. This manuscript provides novel insights in that regard. The prose is clear and overall the paper well-written. The methods are appropriate and well-executed. The results and discussion are reasonable and well-supported, and the figures were clear and helpful in displaying relevant data. I have a few, relatively minor concerns with assumptions that are detailed below. These concerns boil down to: 1. Are all temperate phages integrative? If not, how to better account for lysogeny? 2. The use of multiple bioinformatics tools to resolve some aspect of your data set (e.g., calling viral OTUs) is commendable, but it was not clear how the multiple outputs were resolved into a single "answer" that was then reported out. It would be very helpful to have more details on how these many (and potentially conflicting) outputs were resolved into what the authors regarded as their final results.

Specific Comments

Introduction

l. 37 – 39: the previous sentence described the viral shunt, a mechanism by which viral lysis of cells re-directs carbon and energy flows from higher trophic levels. While the expression of "virus-encoded metabolisms" (viral metabolic genes? As in AMGs?) can be part of overall viral impacts on nutrient cycling, it is not part of the canonical viral shunt.

l.39: a minor point, but it might be more to the point to state "viral metagenomes," as "omics" implies broader scale holistic methods (e.g., proteomics, metabolomics, etc.) and the papers cited are squarely metagenomes.

l. 43-49: a well-reasoned rationale for the system of study (arid soils, wet-up processes).

l. 50: to be clear, is this saying that a greater proportion of cell death has been observed among the Actinobacteria and Proteobacteria, as compared to the total bacterial community? This was a bit confusing.

l. 58: the archetype temperate phage lambda encodes an integrase, and some prophages do integrate into the host chromosome during lysogeny. But some do not. More so than integration/excision, the repression-derepression of lytic functions is a hallmark of temperate phages. I take those integrase gene counts with a grain of salt, as an index of lysogeny.

Methods

l. 479: what was the range of DNA yields for the virus extraction (e.g., per 5g replicate)? Since it is not specified, I assume that whole genome amplification (WGA) approaches were not used prior to sequencing.

l. 485: co-assembly of all replicates per timepoint. I assume this was performed to maximize assembly of contigs, but it does eliminate the ability to describe variation across replicates.

l. 488: "uniport" – should this be Uniprot?

l. 492 and 504: I appreciate the use of multiple algorithms for binning contigs and for calling viral OTUs, respectively. But how were the outputs from the various tools integrated into a what the authors considered the "final" or reported answer? For example, in the case of vOTUs, did a particular set of sequences have to be grouped by all algorithms to be considered "real"? What was done in cases where a set of sequences were grouped by one tool, but some of those sequences were excluded by another tool? How did the authors resolve these differences? Which calls were "best" and how did the authors make that determination? Please include your rationale in the manuscript.

L. 512: I assume integrase was a gene of interest for identifying putative temperate phages. Hopefully there will be some discussion (e.g., in the Discussion section) of the shortcomings of this

approach.

I. 555 – 558: Ok, will be looking to see more explanation on relaxing match criteria to 50%. This seems arbitrary. Why do you think that few vOTU reads could be mapped to your qSIP data?

I. 581-597: assigning putative host taxonomy. As with previous processes, I think the authors do well here to use more than one tool to try to assess the potential identities of viral hosts. I am not clear on how the results between CRISP spacer searches and shared ORF annotations were compared between each other, and how potential conflicts in results between the two methods were resolved. It is mentioned on I. 603 about consensus, but was there consensus in all cases?

L. 619 – 620: I think it's important to consider alternative hypotheses as to why coverage breadth might be less than 100 %.

I. 635: It is worth noting that PropagAtE assumes integrative prophages, and while we assume that integration is part and parcel of lysogeny, there are known examples of temperate phages that do not integrate (e.g., N15 and P1 families). Further, we do not know the extent to which non-integrative phages make up the total population of temperate prophages.

I. 657: I understand why the focus was on integrase. However, this may not be the best diagnostic gene for lysogeny. Is it possible to conduct a similar analysis examining vOTUs that did or did not contain repressor proteins? Particularly, if both analyses support each other then this would increase confidence in the results.

I. 704 – 708: these calculations come with a lot of fudge factors, but very pleased to see the assumptions involved explicitly stated.

Results

I. 103: I think it's just "extracellular"

I. 120: this is a very interesting observation, that patterns of viral OTU presence were not reproducible across replicates – whereas patterns of bacterial OTUs were. Why is this the case? Will be looking for considerations in Discussion.

I. 127: probably best to define how richness was determined in I. 123, when richness is first mentioned in the Results section.

I. 129: is the "number of vOTUs" here different from richness? IF so, please explain what is meant in this context. If not, why not just use "richness"? This is potentially confusing.

I. 147: there are 4 categories, each representing up to 15% of total virome reads... which means that, at best, 40% of reads did not fit into any of these categories. What are we to make of these reads? To me, this suggests that in addition to spatial heterogeneity, there is much stochasticity that may be difficult to characterize or explain when it comes to viral dynamics in soils.

I. 158: this would seem to support your hypothesis regarding KtW. Is it worth pointing this out explicitly? But then I. 162 – perhaps this trend is not generalizable.

I. 165 – 170: here, it is worth establishing the limit of detection for your methods. Do we know what the lower limit of copy number is for a vOTU to be detected? Because if such signatures are present at early time points, but cannot be detected, then this scenario would yield the same outcome (and conclusions). I think it's worth explicitly mentioning LoD here and how it could skew perceptions of results.

L. 191: Interesting! Will be looking for more on this in Discussion.

I. 195 – 206: what do you make of this pattern (or lack of clear pattern)? To me, it suggests that a single host taxon may be producing more than one viral OTU. Again, looking for this in Discussion.

I. 223: see previous comments about limitations of integration as the single defining characteristic of temperate phages. It's a great start, and an accepted method, but it may not catch everything. Why not check these results against classical prophage induction assays (e.g., with mitomycin C)? I am not suggesting this *must* be done, as the manuscript already describes an impressive amount of work. But such parallel approaches would only help in sussing out answers to these complex questions.

I. 235: good analysis to check results on genome assembly.

I. 259: this is an interesting idea, and really shows the authors trying to consider other possibilities besides prophage induction. But in conducting this search specifically for Inovirus-like sequences, why not also look for N15- or P1-like sequences? Or search for representation of repressor proteins in addition to integrase?

Discussion

I. 285: To be clear, I would use language like “obligate lytic phages dominate following wet-up” because “a lytic lifestyle is predominant” can suggest that temperate phages (which can replicate through either lytic or lysogenic mode) are being pushed to lytic mode. And my impression from your data and analysis is that you are arguing that temperate phages are not well-represented in your sample populations, and so it’s obligate lytic phages that are doing most of the activity/response to wet-up.

L. 293: “the virions present at the end of the dry season determined the trajectory of the plot-specific viral composition.” And then I. 297: “These results imply that dry soil may be a reservoir, or seedbank, of diverse viruses, a subset of which becomes dynamic following rewetting.” What we are seeing with wet-up is a reduction in viral richness, and a response & increase of a narrow subset of viruses. An enrichment of a less-diverse population. How does this generate the diverse seedbank that is proposed?

L. 324: Interesting!

I. 336: are temperate phages, identified as integrated elements in MAGs, highly represented in your soils, in general? (i.e., not just during wet-up, but at any time point)?

I. 345: this seems a decent place to mention the existence of temperate phages that do not integrate.

I. 389: it is important to recall your sample handling. Soils were collected from the field, homogenized, and sieved to 2mm. While I understand why this was done, the fact remains that this would obliterate any spatial relationships between host and phage. Why, even after this homogenization step, would bacterial taxa appear to be more-or-less monodisperse, while viral taxa are not? Are different mechanisms available to each group (i.e., bacteria vs. viruses) when it comes to motility, mobility, and forces affecting attachment, desorption, and transport?

L. 402: excellent analysis and question!

I. 410: and, in fact, your data are suggestive that a given vOTU may be able to replicate in more than one host taxon. How else to explain the observed trends?

Figures

Fig. 1 – clear overview of experimental design and approaches, very helpful in conceptualizing.

Fig. 3 – I was looking for more discussion of how reverse L-V dynamics are even possible, since one typically expects an increase in host population FIRST to support an increase in the parasite (viral population). I may have missed where this was mentioned, but to me, this suggests that the addition of water (and perhaps transport of soluble nutrients to cells) enabled stalled or otherwise slow-moving viral infections to rapidly complete, leading to this initial burst of vOTU that precedes any increase in host (e.g., Fig 3B, Actinobacteria panel). The authors are of course free to disagree with this interpretation, but I do think that they should include hypotheses that could explain these unexpected dynamics.

Reviewer #2 (Remarks to the Author):

This manuscript investigated the viral response and virus-host dynamic following wet-up using viromic and metagenomic approach, and estimated the virus induced bacterial mortality using a qSIP approach.

Major comments:

1. One of the major conclusions was that lytic viruses were predominant in the wet-up process of the soil, however, the extracted VLPs were extracellular VLPs, in which lytic viruses could be dominant, and thus resulting in lower recovery of lysogenic viruses and underestimated diversity and abundance. The intracellular VLPs could harbor a higher proportion of lysogenic viruses. Induction of the release of temperate viruses, for example, mitomycin treatment of bacterial fraction, could be useful. Identifying vOTUs from the metagenome could be another option to recover vOTUs, including intracellular ones, that suffered from the low proportion of viral sequences in the metagenome. This could be, at least in part, the reason for some results, for example, late vOTUs were not detected in 0-24 h metagenome (L165-170), post-wet-up detected vOTUs were not significantly enriched in integrase genes (L240)

2.L112-121, Fig 2A, A total of 542 MAGs were obtained, while, only 170 MAGs were included in this analysis, why? Were these MAGs potential hosts? If that so, this should be stated in the figure captions and methods. In addition, since all vOTUs were included (Fig 2A right), I would suggest a plot showing the succession of total bacterial communities, or at least, all MAGs should be included for better comparison.

3. Figure 3C, active vOTUs. The active vOTUs were from metagenome rather than virome, the detailed method for identifying vOTUs from metagenome should be presented. 229 vOTUs were recovered from metagenome (L103), of which 177 were isotope enriched and 58 were detected in virome (L203-204). The idea that using qSIP to identify active vOTUs is great and of ecological importance, while, comparing to the number of identified vOTUs from VLP viromes, such a small fraction of active vOTUs may not be sufficient to characterize the active vOTUs. Figure 3C, isotope-enriched MAGs and vOTUs were included, another question is that whether these MAGs were the host of these vOTUs? or they just represent MAGs from the same lineage and vOTUs potentially infect this lineage? The number of MAGs and vOTUs were presented, how about their abundance?

Minor comments:

1. Fig 2A, It was a bit difficult for me to discriminate samples between 72 and 168 h, I would appreciate if the authors consider change the colors for better reading.
2. L162-163, what was the "viral signatures of host-specific successional dynamics", a bit more explanation could be better for reading.
3. L187-193, so why this happened, a reversed Lotka-Volterra dynamic?
4. L291-293, figure 2 right, it seems that most of later time point samples (48-168h) were distinct from early time point (0-24), did it suggest a more similar viral community structure?
5. Virus-induced mortality increased with time point (Figure 5B), suggesting there should be more virions in soil, but why the viral DNA decreased during 3-48 h (Figure 2 C)?
6. L520-521, 1.450-1.7800 g/ml?, should be 1.750-1.7800 g/ml?
7. L678, ref 49, this is an unpublished data, I am sorry I did not look into the details due to time limit. But maybe some sentences to describe the result would be better for unpublished data.

Reviewer #3 (Remarks to the Author):

In this manuscript, the authors describe the dynamics of soil viral communities following wet up in a microcosm experiment and estimate viral contributions to microbial community turnover. Using deep sequencing and metagenomic assembly, coupled with time series and stable isotope profiling, the authors describe presence and turnover of soil virions, suggesting that temperate phages do not account for the majority of the viruses present and that the soil viral community does not conform to canonical Lotka-Volterra behaviors, and that spatial heterogeneity plays an important role in shaping soil viral communities and their response to perturbation.

Although this work is well written and clearly communicated, it would be improved by the consideration of the following:

1. line 258 -- the methods associated with the search for Inoviridae via HMM search for the pI-like ATPase protein family do not appear to be included in the methods section. Please provide these additional details.
2. line 269-270 -- This sentence describes the highest per-day viral contribution to microbial mortality taking place at 24 hours, but the range of values described (0.02 to ~17%) are lower than those described in the previous sentence (i.e., those for 168 hrs). Please clarify this statement.
3. line 459 -- reference to Life Technologies. Please confirm that full manufacturer information is provided for each kit and/or platform referenced. (e.g., Life Technologies vs. Beckman-Coulter, Indianapolis, IN, USA in line 455).
4. line 488 -- Should this be UniProt instead of Uniprot? Additionally, in conjunction with the bioinformatics tools used, please state whether default parameters were used or if custom

settings, thresholds, etc., were applied.

5. Figure 4B -- please include a description of the error bars in the figure caption.

6. Supplemental Figure 2 -- It appears that the rows of each heat map are independent of the heat maps produced for the other field sites, but it would be helpful to confirm this more explicitly in the figure caption.

7. Supplemental Figure 5B -- Please include a description of the error bars in the figure caption.

Summary Response to Reviewers

We appreciate the thoughtful feedback of the reviewers. Here we summarize our revisions in response to three primary issues raised by all three reviewers: we (1) clarified our use of different approaches throughout the main text and in our methods, (2) explicitly outlined how we integrated different approaches to describe virus response following rewetting, and (3) expanded on our approach and potential biases to estimate lysogenic viruses. Please see the following added text (new text added to our manuscript revision is written in blue throughout our response to reviewers):

Following this summary, we respond directly to each reviewer comment.

1) Clarifying our use of different approaches and 2) how we integrated the different approaches to describe virus response following rewetting

Multiple sequencing approaches (Supplementary Table 1) provided distinct insights into viruses in soil. Metagenomes reveal viruses in multiple life stages: integrated into host chromosomes, virions in host cells, or extracellular virions. Quantitative stable isotope probing (qSIP) shows the continuum of activity of viruses found in metagenomes and their predicted bacterial hosts. Viromes offer a focused view of extracellular viral-like particles (VLPs). Together these techniques enable an understanding of viral dynamics from a total community perspective (metagenomes), from a virion perspective (viromes), and from quantified activity estimates (qSIP).
(Lines 77-84)

To test our hypothesis that wet-up induces temperate viruses we combined our sequencing methods to take a: (1) community sequence-centered approach: cross-mapping of vOTUs to metagenome sequences to look for vOTUs in the context of bacterial sequence; (2) vOTU-centered approach: detection of capacity to integrate into a host genome via integrase genes; (3) MAG-centered: prophage detection in MAGs.
(Lines 220-225)

We unified metagenomics and viromics to generate a library of vOTUs (Figures 2-4). We combined these methods to characterize lysogeny in our system as prophages in MAGs (metagenomes), as integrase encoding vOTUs (metagenomes and viromes), and through cross mapping of viromes and metagenomes (Figure 4). We integrated results from 16S rRNA gene qSIP (active bacteria) and virome DNA (estimate of viral particles) to quantify viral contribution to bacterial turnover (Figure 5). We applied multiple sequence-based approaches to minimize inherent biases of each strategy.
(Lines 308-315)

Our calculation integrated 16S rRNA gene targeted qSIP-estimated microbial mortality rates (most representative of the soil community profile) and virome viral abundances (assaying extracellular viruses to best proxy cell lysis events).
(Lines 285-287)

3) Expanding on our approach and potential biases in estimating lysogenic viruses

Regarding whether our virome sequencing led to an underestimation of the lysogenic virus populations, our findings on lysogeny were not limited to estimating temperate viruses from vOTUs. We used multiple approaches and reasoned that similar conclusions would support reporting that wet-up does not serve as an environmental inducer of lysogens. To underscore our multiple methods, we combined Figure 4 which centered on integrase detecting in vOTUs with Supplementary Figure 6 which visualized prophage detection in MAGs.

We updated the results section with the following text to clarify that rather than leading to an underestimation of lysogenic viruses, looking for temperate viruses in our vOTU set best tests our hypothesis regarding temperate viruses:

Our vOTU set is primarily comprised of vOTUs resolved from virome assemblies, which represent detected extracellular viruses (in a lytic cycle). This approach provided an ideal pool of viruses to test our hypothesis that wet-up serves to induce temperate viruses. (Lines 239-241)

However, we agree with the premise of the reviewer's comment: shown here and by others, recovering 100x more viruses in our viromes than in our metagenomes (Nicolas et al., 2021; Santos-Medellin et al., 2021) has the inherent bias of focusing on viruses in a lytic cycle (or in some chronic infection state that we do not yet have tools to understand well). Metagenome-detected viruses may have the opposite bias – favoring intracellular viruses since it is difficult to discern the state of a metagenome-detected virus unless it is flanked by bacterial sequence i.e. integrated into a host genome. Because metagenomes alone cannot provide an equivalent view into the viral fraction of soil, the set of viruses used here has the biases intrinsic in current soil ecovirology methods.

Reviewers also commented on our use of lysogeny through the manuscript. We were careful in our findings to state that lysogeny did not appear to be prevalent based on our analyses. We did not state the converse: viruses in our system were primarily obligate lytic. We have clarified throughout the text that our study of lysogeny confined us to temperate viruses capable of integration, and not explicitly of the extrachromosomal mechanism of maintenance or phage-plasmids (Pfeifer et al., 2021). We agree with reviewers that each metagenomic method used limited our conclusions on lysogeny during rewetting, and that non-integrating temperate viruses or phage-plasmids, represent promising future research that falls outside the scope of these analyses. It appears no viral hallmark gene has been identified and benchmarked that would facilitate a streamlined analysis of non-integrating temperate viruses.

We have added the adjective “integrating” to delineate temperate viruses more clearly. We rephrased the discussion to be clearer about the analyses we conducted and explained the limitations of our integrase-based analysis. New text is shown bolded and underlined:

Multiple metagenomic methods, individually and comparatively, provided complementary analyses to search for potential integrated viruses and their dynamics following rewetting. Taken together, our data suggests that vOTUs which cannot recombine into a host chromosome through an integrase dominate in this system.
(Lines 362-364)

Further, our methods focused on temperate viruses that may integrate into their host chromosomes. While neither the capacity to integrate into a host chromosome nor persistence as an integrated prophage is prevalent in our wet-up study, we cannot rule out other mechanisms of viral maintenance in a host cell when a virus is not lytically programmed, such as those of non-integrating temperate viruses like phage-plasmids^{27,72} or pseudolysogeny⁷³. Our understanding would benefit from future studies that focus on the prevalence and dynamics of phage-plasmids in microbiomes and soil systems⁷². Supporting the prevalence of non-integrating temperate viruses or some other cell-associated state of soil viruses, the majority (67%) of viral populations detected as active were only found in metagenomes (direct DNA extraction from soil) rather than viromes.
(Lines 381-390)

Similarly, our viral lifestyle analyses limited us to identification of prophage only as insertions in host chromosomes which further constrained our ability to match viruses with their hosts. This precluded analysis and understanding of phage-plasmids which persist as extrachromosomal plasmids.
(Lines 460-463)

Here we see the power of using metagenomes and viromes as complementary analyses: both perspectives corroborated integrating temperate viruses neither predominate in this system nor in the response to soil rewetting.
(Lines 371-374)

Reviewer #1 (Remarks to the Author)
General Comments

This manuscript by Nicolas et al. describes work to characterize the response of bacteria and their associated viruses to wet-up, a phenomenon that occurs when water is added to dry soils. The response of dryland soil bacteria to wet-up has been investigated for decades, with increasing sophistication as methods expand. Very few studies have characterized the response of soil viral populations, nor their interactions with bacterial hosts during this process. This manuscript provides novel insights in that regard. The prose is clear and overall the paper well-written. The methods are appropriate and well-executed. The results and discussion are reasonable and well-supported, and the figures were clear and helpful in displaying relevant data. I have a few, relatively minor concerns with assumptions that are detailed below. These concerns boil down to: 1. Are all temperate phages integrative? If not, how to better account for lysogeny? 2. The use of multiple bioinformatics tools to resolve some aspect of your data set (e.g., calling viral OTUs) is commendable, but it was not clear how the multiple outputs were resolved into a single “answer” that was then reported out. It would be very helpful to have more details on how these many (and potentially conflicting) outputs were resolved into what the authors regarded as their final results.

Thank you for this overall positive assessment of our manuscript. As we explain in our summary response (above), we sought to use multiple bioinformatic tools as a series of complementary perspectives into the soil microbiome. We have added text (lines 289-300, 347-349, 352-361, 444-448) to delineate which results came from specific datasets (e.g., for our analysis of lysogeny) and to clarify when multiple datasets were used for combined analysis (e.g., estimation of virus-induced microbial mortality).

Specific Comments

Introduction

1. *I. 37 – 39: the previous sentence described the viral shunt, a mechanism by which viral lysis of cells re-directs carbon and energy flows from higher trophic levels. While the expression of “virus-encoded metabolisms” (viral metabolic genes? As in AMGs?) can be part of overall viral impacts on nutrient cycling, it is not part of the canonical viral shunt.*

Thank you. We updated the text accordingly and switched the sentence order:

Viral impact on microbiomes and biogeography is thought to occur through targeted predation shaping microbial community composition and via expression of viral-encoded metabolisms (“auxiliary metabolic genes”)^{8–11}. In marine systems, viral predation has been approximated to account for killing upwards of ~40% of bacteria daily and redistributing up to 55% of bacterial carbon via the “viral shunt”^{9,12,13}.
(Lines 35-39)

2. *L.39: a minor point, but it might be more to the point to state “viral metagenomes,” as “omics” implies broader scale holistic methods (e.g., proteomics, metabolomics, etc.) and the papers cited are squarely metagenomes.*

We have changed the wording as recommended.

3. *I. 43-49: a well-reasoned rationale for the system of study (arid soils, wet-up processes).*
Thank you.

4. *I. 50: to be clear, is this saying that a greater proportion of cell death has been observed among the Actinobacteria and Proteobacteria, as compared to the total bacterial community? This was a bit confusing.*

We clarified the text as follows:

Given observed differential mortality for microbial phyla after rewetting, where *Actinobacteria* and *Proteobacteria* exhibited higher mortality rates than other lineages^{19,23}, we hypothesized that ... (Lines 49-51)

5. *I. 58: the archetype temperate phage lambda encodes an integrase, and some prophages do integrate into the host chromosome during lysogeny. But some do not. More so than integration/excision, the repression-derepression of lytic functions is a hallmark of temperate phages. I take those integrase gene counts with a grain of salt, as an index of lysogeny.*

We agree that using integrase genes alone as an index of lysogeny will not give a complete picture of temperate phage lifestyle in soils. For this reason, we applied three orthogonal approaches to quantify lysogeny: 1) we searched for vOTU genomes found in both viromes and metagenomes that contained bacterial flanking regions in metagenomes, 2) we searched for vOTUs that contained an integrase gene in both viromes and metagenomes, and 3) we took a MAG-centric approach, and predicted prophages in the context of bacterial genomes. All three approaches suggest that lysogeny via recombination into the host chromosome is not the dominant lifestyle in our experiment. As noted above, current bioinformatic methods are still limited in their ability to identify phage-plasmids i.e., non-integrating temperate viruses. We have updated the text to make our three approaches clearer and more distinct:

Multiple metagenomic methods, individually and comparatively, provided complementary analyses to search for potential integrated viruses and their dynamics following rewetting. (Lines 362-364)

We applied three orthogonal approaches to quantify lysogeny outlined in our added text, lines 220-225 (Page 1).

We also searched the literature for canonical repression-derepression genes—this has been most well-studied in lambda-like phages (i.e., the lytic repressor, *cI*). We did not find a systematic study of temperate viruses to discern a temperate “pangenome” across reference databases or environments. It appears that viral integrases are a diverse family of recombinases that may be more widely distributed than lytic repression-derepression genes. In response to comment 16 we looked for repressors in our vOTU set and found that this annotation is not as widespread as integrases.

Methods

6. *I. 479: what was the range of DNA yields for the virus extraction (e.g., per 5g replicate)? Since it is not specified, I assume that whole genome amplification (WGA) approaches were not used prior to sequencing.*

We have added a metadata spreadsheet (Supplemental Table 2) which includes per sample DNA extracted, total reads generated, and other relevant data. In our 10g soil virus extractions,

we had yields ranging from 0.663 ng DNA (3 hours) to 8.411 ng DNA (168 hours). We added the following text to the methods:

DNA was sent to the DNA Technologies and Expression Analysis Cores at the University of California Davis Genome Center for library preparation and sequencing. The Swift Accel-NGS 1S Plus DNA Library Kit (Swift BioSciences, Ann Arbor, MI, USA) was used to prepare viromes for sequencing on an Illumina Novaseq (2x150 cycles). Due to the low DNA extraction yields, thirteen rounds of PCR amplification were performed using the Swift unique dual index primers, producing libraries averaging 1 ng/μL. [...] Previous research has shown that for fewer than 14 rounds of PCR amplification standard assembly and analysis pipelines can be used (Roux et al., 2019).
(Lines 528-538)

7. *I. 485: co-assembly of all replicates per timepoint. I assume this was performed to maximize assembly of contigs, but it does eliminate the ability to describe variation across replicates.*

Coassembly could minimize sequence variation across replicates. However, we assembled sequences, established a set of genomes and then dereplicated these genomes such that coassembly likely serves as a dereplication step at the assembly stage. Specifically, a sequence will be assembled fewer times per timepoint and therefore, given differential coverage across replicates per time point, we likely attain a better assembly with coassemblies (Greenlon et al., 2022). To assess variation across replicates, we mapped individual replicate sample reads back to the total set of MAGs or viral genomes (lines 612-626, 619-623); this allowed us to capture genome coverage per replicate per time point. This is how we established error bars in Figure 4.

We added the following text to our methods section to clarify this reasoning:

Co-assembly of replicate samples may serve as an additional dereplication step at the assembly stage because a consensus sequence will be assembled fewer times per timepoint. Given differential coverage across replicates per time point, co-assemblies can generate better assemblies (Greenlon et al., 2022).
(Lines 539-542)

8. *I. 488: “uniport” – should this be Uniprot?*

Yes, we corrected the text.

9. *I. 492 and 504: I appreciate the use of multiple algorithms for binning contigs and for calling viral OTUs, respectively. But how were the outputs from the various tools integrated into a what the authors considered the “final” or reported answer? For example, in the case of vOTUs, did a particular set of sequences have to be grouped by all algorithms to be considered “real”? What was done in cases where a set of sequences were grouped by one tool, but some of those sequences were excluded by another tool? How did the authors resolve these differences? Which calls were “best” and how did the authors make that determination? Please include your rationale in the manuscript.*

For vOTUs (updated text 558-570) we describe the tools we used to predict viral sequences from both viromes and metagenomes. In summary, we subset the viral sequences to only those viruses predicted by at least two of the five bioinformatic tools used. We have updated the text to explain our rationale as follows:

To most robustly establish a viral set we chose to subset these predicted viral sequences to only those identified by more than one tool (Wu et al., 2023) – 777,725 contigs met this criteria, and 32,660 of these sequences were either predicted to be circularized contigs by VRCA (Crits-Christoph et al., 2016) (i.e. implied to be a complete genome), or $\geq 10,000$ base pairs long.
(Lines 564-568)

10. *L. 512: I assume integrase was a gene of interest for identifying putative temperate phages. Hopefully there will be some discussion (e.g., in the Discussion section) of the shortcomings of this approach.*

We acknowledge that using integrases as our primary approach to investigate lysogeny in our vOTU-centric approach largely excludes non-integrating temperate viruses, such as those also known as phage-plasmids (P1, N15, etc.), and we have constrained our conclusions to note this (see response to the overall comments on p. 2). An integrase analysis of vOTUs has caveats but is robust in concert with our parallel methods for assessing lysogeny. We recognize that we cannot capture all temperate viral diversity with a single gene, however this approach has successfully been used before. We added this caveat in our methods section:

All vOTUs were annotated using DRAM-v.¹⁰⁰ To assess viral lifestyle we searched DRAM-v output for integrase-annotated ORFs and denoted integrase-containing vOTUs as temperate viruses with the capacity to integrate into a host chromosome. A single gene approach will not capture all temperate viral diversity, but integrase genes have been used as an effective hallmark with much higher recovery than other genes (e.g., excisionases) in previous peer-reviewed studies^{27,34,55,101–103}.
(Lines 570-575)

11. *I. 555 – 558: Ok, will be looking to see more explanation on relaxing match criteria to 50%. This seems arbitrary. Why do you think that few vOTU reads could be mapped to your qSIP data?*

We added Supplementary Figure 2 which shows the relationship between breadth cutoffs and the number of microcosms a vOTU was present in. We find that the relaxed breadth parameter is robust when applied with the requirement that a vOTU appear in all three replicates for downstream analysis. We observed that only 1,820 total unique vOTUs had mapped reads at the high stringency mapping ID of 98% and with 50% breadth from any time point, SIP density fraction, or plot. Of these 1,820 vOTUs that had reads map at 50% breadth from SIP density fractionated samples (less than 2% of total vOTUs), only 300 vOTUs were detected in all three replicates per time point, evidencing that any mapped reads serves as a bottleneck in detecting active vOTUs from SIP density fractions, and the requirement of appearing in all three replicates of a treatment serves as an additional filter to ensure stringency in our analyses. We added the following text to our methods:

We found that few vOTUs had reads that mapped stringently (98% identity) and were found in all three replicates of multiple density fractions in both ¹⁸O and ¹⁶O treatments (Supplementary Figure 2). Thus, we relaxed the breadth cutoff to 50% as it appeared to be an inflection point between breadth and total vOTUs mapping which balanced between falsely excluding vOTUs that appeared in triplicate in multiple density fractions at a high mapping ID, and falsely including erroneously detected present vOTUs (Supplementary Figure 2). vOTUs meeting this lower breadth cutoff, 50%, in triplicate for a given sample were treated as present in SIP fractions and were analyzed for activity with qSIP calculations (described below).
(Lines 626-634)

We believe that few vOTU reads mapped to qSIP data because the majority of vOTUs are from viromes whereas qSIP data were from fractionated metagenomes, and there was little overlap in these two datasets.

12. *I. 581-597: assigning putative host taxonomy. As with previous processes, I think the authors do well here to use more than one tool to try to assess the potential identities of viral hosts. I am not clear on how the results between CRISP spacer searches and shared ORF annotations were compared between each other, and how potential conflicts in results between the two methods were resolved. It is mentioned on I. 603 about consensus, but was there consensus in all cases?*

For this study, we did not seek to resolve potential conflicts in host prediction since both host prediction methods we used are published and accepted approaches. The new element here was conservatively integrating these host predictions to capture consensus host predictions. We think that predicting host taxonomy or investigations into specific MAG-vOTU linkages would need to be the subject of an entirely different study, see work by a subset of coauthors (Lee et al., 2021). We added the following text:

We used this database to determine the consensus taxonomy between these sources where there were multiple host predictions closest to a species level by searching for the greatest common denominator level of taxonomy. If consensus was not found between host predictions at any level of taxonomy, host taxonomy would be assigned as “unknown.” As the majority of vOTUs neither contained matched spacers nor protospacers, most vOTUs contained only a single host prediction from Kaiju, which was retained.

(Lines 678-682)

13. *L. 619 – 620: I think it’s important to consider alternative hypotheses as to why coverage breadth might be less than 100 %.*

If VLP reads mapped to a subset of a given metagenome-assembled sequence this could also represent fragmentation of a larger sequence that, for whatever reason, assembled as a longer contig from metagenomic sequencing. However, we believe that breadth < 100% here is biologically important: if this was simply about higher quality breadth coverage of viruses in metagenomes then we would have captured this virus in our earlier prediction of vOTUs from metagenomes, and during dereplication this longer vOTU sequence would have become the clustered representative. This initial mapping and breadth cutoff for sequences was used as a tool and was the first in several steps for identifying candidate lysogens through comparative metagenomics. The final step involved manual curation of all potential lysogen candidates.

14. *I. 635: It is worth noting that PropagAtE assumes integrative prophages, and while we assume that integration is part and parcel of lysogeny, there are known examples of temperate phages that do not integrate (e.g., N15 and P1 families). Further, we do not know the extent to which non-integrative phages make up the total population of temperate prophages.*

PropagAtE, and the current suite of bioinformatic tools has not yet accounted for non-integrating prophages. Similar to our response to comment 10, we acknowledge that our MAG-centric analysis of prophages is restricted to prophages that would integrate into MAGs and thus likely excludes non-integrating temperate viruses / phage-plasmids (P1, N15, etc.), and we constrain our conclusions to note this. We believe an integrated prophage analysis of MAGs has caveats

but is robust alongside our parallel methods for assessing lysogeny. We will not capture all lysogeny diversity with this approach, however others have previously used these assumptions with success.

15. *I. 657: I understand why the focus was on integrase. However, this may not be the best diagnostic gene for lysogeny. Is it possible to conduct a similar analysis examining vOTUs that did or did not contain repressor proteins? Particularly, if both analyses support each other then this would increase confidence in the results.*

Thank you for this suggestion. We searched our DRAM-v annotations for *ci*-type repressors, canonical in lambda-like phages, which encompass the non-integrating P1 (circular) and N15 (linear) temperate phages discussed in other comments by reviewer #1. From pfam hits we identified 313 vOTUs containing ORFs annotated as the *ci* repressor (as opposed to generically annotated transcriptional repressors), which determines if transcription is in a lytic or lysogenic mode for viruses. Of vOTUs containing this repressor annotation, 56% also had an ORF annotated as an integrase. We wrote that integrase-containing vOTUs were a minority, 17%, of total vOTUs. Though this is a cursory view of a single well-researched lytic repressor, we see that these vOTUs represent a minor fraction of total vOTUs (~1.2%). Of these repressor-containing vOTUs that do not also encode an integrase, 6 were predicted to be circular and 132 did not circularize. A future study would be needed to ascertain whether these 138 vOTUs (and others) could represent non-integrating temperate viruses. Based on these limited results, we do not find this analysis of the canonical *ci* repressor directly supports or contradicts our stated findings on lysogeny. Because we only found 138 vOTUs out of over 26,000 vOTUs contained this annotation and not also an integrase, it appears to be quantitatively insignificant in our system at the scale of integrases. Since it is unclear how to interpret these results, we opted not to include these findings in our study.

16. *I. 704 – 708: these calculations come with a lot of fudge factors, but very pleased to see the assumptions involved explicitly stated.*

Thank you for the positive feedback.

Results

17. *I. 103: I think it's just "extracellular"*

Corrected as suggested.

18. *I. 120: this is a very interesting observation, that patterns of viral OTU presence were not reproducible across replicates – whereas patterns of bacterial OTUs were. Why is this the case? Will be looking for considerations in Discussion.*

See line 441-447 of the Discussion.

19. *I. 127: probably best to define how richness was determined in I. 123, when richness is first mentioned in the Results section.*

Corrected:

Following rewetting, we found that viral richness, as measured through the number of unique vOTUs per time point, significantly decreased, while viral biomass significantly increased.

(Lines 140-141)

20. *I. 129: is the “number of vOTUs” here different from richness? IF so, please explain what is meant in this context. If not, why not just use “richness”? This is potentially confusing.*

Corrected to read as “richness”.

21. *I. 147: there are 4 categories, each representing up to 15% of total virome reads... which means that, at best, 40% of reads did not fit into any of these categories. What are we to make of these reads? To me, this suggests that in addition to spatial heterogeneity, there is much stochasticity that may be difficult to characterize or explain when it comes to viral dynamics in soils.*

Overall, about 40% of total reads generated in the viromes map to the total viral set through time – 24 hours is an outlier (about 25% of total reads map to the viral set), but at 24 hours we also saw a bloom of candidate phyla radiation (CPR) bacteria. Viromes are viral-enriched metagenomes, but there are other sequences that comprise these samples that we have not investigated as part of the scope of this project (Nicolas et al., 2021). We added the following line to the Results section:

On average, 35.3 % (± 7.5 % standard deviation) of virome reads and 0.55 % (± 0.05 % standard deviation) of metagenome reads mapped to the total vOTU set.
(Lines 119-120)

22. *I. 158: this would seem to support your hypothesis regarding KtW. Is it worth pointing this out explicitly? But then I. 162 – perhaps this trend is not generalizable.*

This is a possible interpretation which we discussed in original lines **417-421**, but did not discuss this result as “kill the winner” directly because it marks an observed feature of the system and not necessarily a consistent temporal dynamic when looking at viral and microbial activity.

23. *I. 165 – 170: here, it is worth establishing the limit of detection for your methods. Do we know what the lower limit of copy number is for a vOTU to be detected? Because if such signatures are present at early time points, but cannot be detected, then this scenario would yield the same outcome (and conclusions). I think it’s worth explicitly mentioning LoD here and how it could skew perceptions of results.*

We removed the parts of the results and discussion section that speculate that “late vOTUs” appearing in metagenomes at earlier time points could be evidence of temperate viruses. As discussed in other review responses, this is a much more complicated question that could serve as the basis for the next study to emerge from these data.

Preliminary analysis of the minimum relative abundance of a vOTU in the virome compared to the metagenome suggests the same sensitivity in detection. This does not account for sequencing rarefaction – viromes on average generated 73 Gbp of sequence per sample and unfractionated metagenomes generated on average 23 Gbp of sequence per sample. We have added this information to the results section:

We found that vOTUs in the metagenome and in the virome showed a similar minimum detected relative abundance on the order of the 10^{-8} (viromes: 8.2×10^{-8} , metagenomes: 4.79×10^{-8}).
(Lines 114-116)

24. *L. 191: Interesting! Will be looking for more on this in Discussion.*

Thank you for the positive feedback.

25. *I. 195 – 206: what do you make of this pattern (or lack of clear pattern)? To me, it suggests that a single host taxon may be producing more than one viral OTU. Again, looking for this in Discussion.*

See lines 449-452 of the Discussion.

26. *I. 223: see previous comments about limitations of integration as the single defining characteristic of temperate phages. It's a great start, and an accepted method, but it may not catch everything. Why not check these results against classical prophage induction assays (e.g., with mitomycin C)? I am not suggesting this *must* be done, as the manuscript already describes an impressive amount of work. But such parallel approaches would only help in sussing out answers to these complex questions.*

We agree. We hope others follow up on studies of induction not just of cultivated soil bacteria (Ghosh et al., 2009), but in soil microcosms and or soil enrichments.

27. *I. 235: good analysis to check results on genome assembly.*

Thank you for the positive feedback.

28. *I. 259: this is an interesting idea, and really shows the authors trying to consider other possibilities besides prophage induction. But in conducting this search specifically for Inovirus-like sequences, why not also look for N15- or P1-like sequences? Or search for representation of repressor proteins in addition to integrase?*

We addressed the questions and concerns here in our response above to the reviewer's comment about *I. 657* (comments 5 and 15).

Discussion

29. *I. 285: To be clear, I would use language like "obligate lytic phages dominate following wet-up" because "a lytic lifestyle is predominant" can suggest that temperate phages (which can replicate through either lytic or lysogenic mode) are being pushed to lytic mode. And my impression from your data and analysis is that you are arguing that temperate phages are not well-represented in your sample populations, and so it's obligate lytic phages that are doing most of the activity/response to wet-up.*

Our original text was conservative to account for non-integrative temperate phage/pseudolysogeny and other viruses that could opt into a lytic cycle that our analysis was not able to pick up. We prefer to keep the original language to remain conservative in our conclusions.

30. *L. 293: "the virions present at the end of the dry season determined the trajectory of the plot-specific viral composition." And then I. 297: "These results imply that dry soil may be a reservoir, or seedbank, of diverse viruses, a subset of which becomes dynamic following rewetting." What we are seeing with wet-up is a reduction in viral richness, and a response*

& increase of a narrow subset of viruses. An enrichment of a less-diverse population. How does this generate the diverse seedbank that is proposed?

In our proposed model, the “seedbank” of rich virions is what survives the dry-down and the wet-up acts as an abiotic selection to enrich for a subset of these virions in dry soil. Our group’s working hypothesis is that as environmental conditions change through the growing season (following the first rain event, simulated in our experiment), that more diverse viruses replicate and become extracellular virions that then persist throughout the dry season when microbial activity is at a minimal, thus generating the observed richness. We added the following text to the discussion for clarity:

Following the reduction in viral diversity observed during the first week after rewetting, we hypothesize that the microbial succession that occurs throughout the growing season⁶⁷ leads to an increase in soil viral richness and that these viruses persist as virions during the prolonged dry season when microbial activity is at a minimum. A previous study similarly observed that dryer soils contained more viral clusters⁶⁸.
(Lines 328-332)

31. *L. 324: Interesting!*

Thank you for this positive feedback.

32. *I. 336: are temperate phages, identified as integrated elements in MAGs, highly represented in your soils, in general? (i.e., not just during wet-up, but at any time point)?*

The answer is unclear. There are still relatively few published genome-resolved soil metagenomic papers and fewer that look at viruses and microbes. Of those at our field site (though sampled at a different location from the soils used in this study), analysis was limited to circularized viruses rather than integrated viruses. Only ten circularized phages were found in the context of ¹³C-SIP density-fractionated metagenomes (Starr et al., 2021). Another study, from the growing season at our site, (Greenlon et al., 2022) does not discuss incidence of integrated phages in MAGs, but examined active viruses during the start of the growing season.

We added the following text to the discussion:

Previous metagenomic studies at this field site^{40,72} have not investigated viral lifestyle, and in soil microbiomes at-large, understanding viral lifestyle is understudied.
(Lines 374-376)

33. *I. 345: this seems a decent place to mention the existence of temperate phages that do not integrate.*

Text has been updated throughout delineating between integrating and non-integrating temperate viruses. We also added the following text to this suggested section of the discussion (see page 2):

Further, our methods focused on temperate viruses that may integrate into their host chromosomes. While neither the capacity to integrate into a host chromosome nor persistence as an integrated prophage is prevalent in our wet-up study, we cannot rule out other mechanisms of viral maintenance in a host cell when a virus is not lytically programmed, such as those of non-integrating temperate viruses like phage-plasmids^{27,72} or pseudolysogeny⁷³. Our understanding would benefit from future studies that focus on the prevalence and dynamics of phage-plasmids in microbiomes and soil systems⁷². Supporting the prevalence of non-integrating

temperate viruses or some other cell-associated state of soil viruses, the majority (67%) of viral populations detected as active were only found in metagenomes (direct DNA extraction from soil) rather than viromes.
(Lines 381-390)

34. *I. 389: it is important to recall your sample handling. Soils were collected from the field, homogenized, and sieved to 2mm. While I understand why this was done, the fact remains that this would obliterate any spatial relationships between host and phage. Why, even after this homogenization step, would bacterial taxa appear to be more-or-less monodisperse, while viral taxa are not? Are different mechanisms available to each group (i.e., bacteria vs. viruses) when it comes to motility, mobility, and forces affecting attachment, desorption, and transport?*

Homogenization was done for soil from each field plot and field plots were kept separate. Thus, despite homogenization we would still be working with dispersal limitations present at our field site. We chose not to mix and homogenize soil from the different field plots to maintain diversity found in plot replicates. We updated the text to clarify this:

The dry soil was transferred to Lawrence Livermore National Laboratory (LLNL) where soil collected from each field plot was individually homogenized and sieved (2mm) to remove large rocks and roots.
(Lines 496-498)

5 g sieved soil from each of 3 field biological replicate plots was weighed into separate 15 mL Nalgene flat bottom vials.
(Lines 507-509)

35. *L. 402: excellent analysis and question!*

Thank you for this positive feedback.

36. *I. 410: and, in fact, your data are suggestive that a given vOTU may be able to replicate in more than one host taxon. How else to explain the observed trends?*

Agreed, we were a bit overly cautious here. The text was revised to state:

we hypothesize that microbes in soil are widely susceptible to infection by many distinct viral populations. However, the lack of specific host-virus linkages in our system prevented investigation of both viral host range and host susceptibility to multiple viruses.
(Lines 451-453)

Figures

37. *Fig. 1 – clear overview of experimental design and approaches, very helpful in conceptualizing.*

We appreciate your comment.

38. *Fig. 3 – I was looking for more discussion of how reverse L-V dynamics are even possible, since one typically expects an increase in host population FIRST to support an increase in the parasite (viral population). I may have missed where this was mentioned, but to me, this suggests that the addition of water (and perhaps transport of soluble nutrients to cells)*

enabled stalled or otherwise slow-moving viral infections to rapidly complete, leading to this initial burst of vOTU that precedes any increase in host (e.g., Fig 3B, Actinobacteria panel). The authors are of course free to disagree with this interpretation, but I do think that they should include hypotheses that could explain these unexpected dynamics.

Multiple mechanisms are possible for reversed L-V dynamics that should be further investigated. We explain in line 344-347 that as described in (Cortez & Weitz, 2014) reversed L-V dynamics are suggestive of observed co-evolution, we have expanded on this in the text as follows:

In contrast, Actinobacteria-infecting viruses appear to show a reversed Lotka-Volterra relationship which is suggestive of coevolution driven dynamics⁵³. In this scenario, viral activity is initially higher than host activity indicating that Actinobacteria are highly susceptible to viral infection – which selects for viral-resistant hosts as demonstrated by Actinobacteria activity increasing after their viruses.
(Lines 344-347)

Reviewer #2 (Remarks to the Author):*Major comments:*

1. *One of the major conclusions was that lytic viruses were predominant in the wet-up process of the soil, however, the extracted VLPs were extracellular VLPs, in which lytic viruses could be dominant, and thus resulting in lower recovery of lysogenic viruses and underestimated diversity and abundance.*

Viromes gave us a uniquely “lytic” view into the studied soil system. However, extracellular VLPs also capture temperate viruses in their lytic lifestyle. Currently, generating viromic sequence data represents the best method in the field for establishing a set of diverse viruses as many more viruses are recovered in the virome compared to the metagenome (Santos-Medellin et al., 2021). Further, we used several methods to detect lysogenic viruses and each showed integrated prophages were a minor proportion of total viruses (lines 362-374). Please also see pages 2-3 and the response to reviewer #1 comment 33.

2. *The intracellular VLPs could harbor a higher proportion of lysogenic viruses. Induction of the release of temperate viruses, for example, mitomycin treatment of bacterial fraction, could be useful. Identifying vOTUs from the metagenome could be another option to recover vOTUs, including intracellular ones, that suffered from the low proportion of viral sequences in the metagenome. This could be, at least in part, the reason for some results, for example, late vOTUs were not detected in 0-24 h metagenome (L165-170), post-wet-up detected vOTUs were not significantly enriched in integrase genes (L240)*

We have clarified in the text that our set of viruses includes extracellular vOTUs and metagenome-detected (potentially intracellular) vOTUs:

The total set of 26,368 dereplicated vOTUs in this study represents 229 vOTUs resolved from metagenomes and 26,139 assembled in the virome (Figure 1). (Lines 112-114)

Please also see lines 116-119:

We consider vOTUs detected in viromes to be extracellular VLPs, which are distinct from vOTUs detected in metagenomes where analysis cannot differentiate between intracellular and extracellular viral populations (Supplementary Table 1).

Since the vast majority of vOTUs detected in this system are from the virome, we agree that we could be missing intracellular viruses with our analyses. However, given the differences in recovery of viruses in metagenomes vs. viromes (discussed above), we believe our analyses represent the most current and comprehensive view of soil viruses – most studies that discuss soil viruses are limited to mining viruses from metagenomes and thus vastly undersample viral diversity or only generate viromes and have no cellular fraction as a reference point. We also discuss that we identified viruses in the context of MAGs using multiple methods (lines 689-721).

3. *L112-121, Fig 2A, A total of 542 MAGs were obtained, while, only 170 MAGs were included in this analysis, why? Were these MAGs potential hosts? If that so, this should be stated in the figure captions and methods. In addition, since all vOTUs were included (Fig 2A right), I would suggest a plot showing the succession of total bacterial communities, or at least, all MAGs should be included for better comparison.*

We clarified the text and updated Figure 1 to be more precise:

We resolved 377 MAGs ($\geq 50\%$ complete, $\leq 10\%$ contamination) which we dereplicated to 338 MAGs⁴⁹ and combined with 168 dereplicated MAGs from a previous study at our field site which sampled soil during the previous winter⁴⁰. In total, we studied a combined set of 542 dereplicated MAGs: [...]

(Lines 106-109)

We further clarified why 170 MAGS were included in Figure 2A:

Of our total dereplicated MAGs, 170 met breadth criteria to be considered present during the first week following rewetting.

(Lines 129-130)

4. *Figure 3C, active vOTUs. The active vOTUs were from metagenome rather than virome, the detailed method for identifying vOTUs from metagenome should be presented.*

We used the same methods for identifying vOTUs from metagenomes as we did for viromes, see the section of methods beginning line 558.

In reviewing methods, we added details on assessing prophages in MAGs:

Identified prophage-containing MAGs were manually matched from their GTDB to a NCBI taxonomy (to align with vOTU host match names). We established the count of MAGs per phylum that contained or did not contain a prophage. We summed the relative abundance of MAGs by whether the MAG had a detected prophage to generate a seaborn lineplot overlaid with a scatterplot of relative abundances split into these categories.

(Lines 716-721)

5. *229 vOTUs were recovered from metagenome (L103), of which 177 were isotope enriched and 58 were detected in virome (L203-204). The idea that using qSIP to identify active vOTUs is great and of ecological importance, while, comparing to the number of identified vOTUs from VLP viromes, such a small fraction of active vOTUs may not be sufficient to characterize the active vOTUs. Figure 3C, isotope-enriched MAGs and vOTUs were included, another question is that whether these MAGs were the host of these vOTUs? or they just represent MAGs from the same lineage and vOTUs potentially infect this lineage? The number of MAGs and vOTUs were presented, how about their abundance?*

The detailed methods for identifying vOTUs in metagenomes are presented in lines 558-563.

We believe this is the first study to overlay SIP-density fractionated metagenomes onto viromes. In general, very few studies pair viromes and metagenomes, especially in soil. It is still unclear which approach provides the most holistic characterization of active vOTUs (i.e. viromes, viruses from metagenomes, SIP-metagenomes, SIP-viromes, etc.), but we believe that identification of SIP active viruses from metagenomes provides important information, and we were conservative in all of our estimates and conclusions due to the need for future studies to better define biases/viewpoints of the different metagenomic datasets. Please also see the discussion of viruses in viromes compared to metagenomes in response to reviewer 1 comment 12 (l. 555 - 558).

In Figure 3C we are using MAGs and vOTUs of the same lineage, these are not necessarily the specific host genomes of these viruses. We added text to clarify:

The number of detected active MAGs and vOTUs of the same lineages followed similar temporal trends for all three phyla we focused on.

(Lines 204-205)

Minor comments:

6. *Fig 2A, It was a bit difficult for me to discriminate samples between 72 and 168 h, I would appreciate if the authors consider change the colors for better reading.*

We changed the colors to be more discernible and we used a colorblind friendly palette.

7. *2.L162-163, what was the “viral signatures of host-specific successional dynamics”, a bit more explanation could be better for reading.*

We changed this sentence to read:

Overall, we found that vOTUs of prevalent hosts in our system (Actinobacteria and Proteobacteria) appeared as the most prevalent vOTUs in each response category. However, host-specific successional dynamics previously detected of bacteria following wet-up^{20,23} were not reflected in our viral host predictions.
(Line 177-80)

8. *L187-193, so why this happened, a reversed Lotka-Volterra dynamic?*

Yes, please see our response to reviewer 1 comment 38.

9. *L291-293, figure 2 right, it seems that most of later time point samples (48-168h) were distinct from early time point (0-24), did it suggest a more similar viral community structure?*

In an earlier manuscript draft we discussed “early” vs. “late” in the wet-up time course, but since we did not further investigate these early vs. late clusters due to text limitations, we prefer to leave this discussion point out.

10. *Virus-induced mortality increased with time point (Figure 5B), suggesting there should be more virions in soil, but why the viral DNA decreased during 3-48 h (Figure 2 C)?*

Figure 5B shows cumulative mortality through time – we added “cumulative” to the figure legend to clarify. Our contribution to mortality is a percentage based on number of virions scaled to the number of lysed microbial cells over the calculated cumulative total microbial mortality (equation 2). When DNA decreases 3 - 48 h (perhaps insignificantly), the rate of microbial mortality decreases during this timeframe, thus cumulative viral mediated mortality still increases though the number of virions in soil proxied by DNA may not increase.

11. *L520-521, 1.450-1.7800 g/ml?, should be 1.750-1.7800 g/ml?*

We have made the following correction:

5 groups based on density (1.6400-1.7039 g/ml, 1.7040-1.7169 g/ml, 1.7170-1.7299 g/ml, 1.7300-1.7449 g/ml, 1.7450-1.7800 g/ml).

12. *L678, ref 49, this is an unpublished data, I am sorry I did not look into the details due to time limit. But maybe some sentences to describe the result would be better for unpublished data.*

We added the following text for further clarification on methods used to calculate the rate of mortality:

Cumulative bacterial mortality was calculated using qSIP estimated mortality rates of 16S rRNA gene sequence and qPCR data as described in Blazewicz et al. 2020²³. In brief, 16S rRNA genes were sequenced from 330 SIP density fractions using primers 515 F and 806 R, as described in Sieradzki et al⁴⁹. Each reaction contained 1 μ L sample and 9 μ L of Phusion Hot Start II High Fidelity master mix (Thermo Fisher Scientific, Waltham, MA, USA) including 1.5 mM additional MgCl₂. PCR conditions were 95° C for 2 min followed by 20 cycles of 95° C for 30 S, 64.5° C for 30 S, and 72° C for 15 S. The triplicate PCR products were then pooled and diluted 10X and used as a template in a subsequent dual indexing reaction that used the same primers including the Illumina flowcell adaptor sequences and 8-nucleotide Golay barcodes (15 cycles identical to initial amplification conditions). Resulting amplicons were purified with AMPure XP magnetic beads (Beckman-Coulter, Indianapolis, IN, USA) and quantified using the Quant-iT PicoGreen dsDNA Assay Kit (Thermo Fisher Scientific) on a BioTek Synergy HT plate reader (Agilent, Santa Clara, CA, USA). Samples were pooled at equivalent concentrations, purified with the AMPure XP beads, and quantified using the KAPA SYBR FAST qPCR kit (KAPA Biosystems, Cape Town, South Africa). Libraries were sequenced on an Illumina MiSeq instrument at Northern Arizona University's Genetics Core Facility using a 300-cycle v2 reagent kit. (Lines 763-779)

Reviewer #3 (Remarks to the Author):

In this manuscript, the authors describe the dynamics of soil viral communities following wet up in a microcosm experiment and estimate viral contributions to microbial community turnover. Using deep sequencing and metagenomic assembly, coupled with time series and stable isotope profiling, the authors describe presence and turnover of soil virions, suggesting that temperate phages do not account for the majority of the viruses present and that the soil viral community does not conform to canonical Lotka-Volterra behaviors, and that spatial heterogeneity plays an important role in shaping soil viral communities and their response to perturbation.

Although this work is well written and clearly communicated, it would be improved by the consideration of the following:

1. *line 258 -- the methods associated with the search for Inoviridae via HMM search for the pl-like ATPase protein family do not appear to be included in the methods section. Please provide these additional details.*

We added the following methods:

To identify potential Inoviridae in our vOTUs we performed HMM searches of known pl-like ATPase protein family using 32 protein models constructed based on alignments of pl-like ATPase from Inoviridae identified in RefSeq, publicly available (https://github.com/simroux/Inovirus/blob/master/Inovirus_classifier/Ino_classifier_db/pl_PC_db_db_annot.hmm). We used HMMER v3.1b2 (<http://hmmer.org>) with default parameters i.e., e-value ≤ 10 . No viral proteins met these lax parameters to be considered homologous to Inoviridae pl-like ATPases.
(Lines 575-582)

2. *line 269-270 -- This sentence describes the highest per-day viral contribution to microbial mortality taking place at 24 hours, but the range of values described (0.02 to ~17%) are lower than those described in the previous sentence (i.e., those for 168 hrs). Please clarify this statement.*

We changed this statement to the following: “Taken cumulatively, **by 168 hours...**” to clarify that the metric reported at 168 hours marks the cumulative mortality occurring throughout the week following wet-up.

3. *line 459 -- reference to Life Technologies. Please confirm that full manufacturer information is provided for each kit and/or platform referenced. (e.g., Life Technologies vs. Beckman-Coulter, Indianapolis, IN, USA in line 455).*

We changed the text to read: “... amount of DNA (Quant-It PicoGreen dsDNA assay (Thermo Fisher Scientific, Eugene, OR, USA))

4. *line 488 -- Should this be UniProt instead of Uniprot? Additionally, in conjunction with the bioinformatics tools used, please state whether default parameters were used or if custom settings, thresholds, etc., were applied.*

Fixed and amended in the text. Methods have been updated with parameter settings.

5. *Figure 4B -- please include a description of the error bars in the figure caption.*

The error bars in Figure 4B represent standard error and have been corrected in Figure 4 and in the associated Supplementary Figure 5. Thank you for this correction.

6. *Supplemental Figure 2 -- It appears that the rows of each heat map are independent of the heat maps produced for the other field sites, but it would be helpful to confirm this more explicitly in the figure caption.*

Yes, because each plot showed a unique set of vOTUs – very few were shared across timepoints and plots – and because these heatmaps are hierarchically clustered based on vOTU relative abundance (i.e. the row order is determined by cluster) there is no correspondence across plot-specific heatmaps between row and vOTU. This is now clarified in the figure legend with the addition of the parenthetical phrase:

Each heatmap corresponds to one of the three field plots sampled and is hierarchically clustered according to vOTU relative abundance (rows are not shared between heatmaps).

7. *Supplemental Figure 5B -- Please include a description of the error bars in the figure caption.*

Thank you, we have added the error bar description to the legend.

References

- Cortez, M. H., & Weitz, J. S. (2014). Coevolution can reverse predator–prey cycles. *Proceedings of the National Academy of Sciences*, *111*(20), 7486–7491. <https://doi.org/10.1073/pnas.1317693111>
- Crits-Christoph, A., Gelsinger, D. R., Ma, B., Wierzos, J., Ravel, J., Davila, A., Casero, M. C., & DiRuggiero, J. (2016). Functional interactions of archaea, bacteria and viruses in a hypersaline endolithic community. *Environmental Microbiology*, *18*(6), 2064–2077. <https://doi.org/10.1111/1462-2920.13259>
- Ghosh, D., Roy, K., Williamson, K. E., Srinivasiah, S., Wommack, K. E., & Radosevich, M. (2009). Acyl-homoserine lactones can induce virus production in lysogenic bacteria: An alternative paradigm for prophage induction. *Applied and Environmental Microbiology*, *75*(22), 7142–7152. <https://doi.org/10.1128/AEM.00950-09>
- Greenlon, A., Sieradzki, E., Zablocki, O., Koch, B. J., Foley, M. M., Kimbrel, J. A., Hungate, B. A., Blazewicz, S. J., Nuccio, E. E., Sun, C. L., Chew, A., Mancilla, C.-J., Sullivan, M. B., Firestone, M., Pett-Ridge, J., & Banfield, J. F. (2022). Quantitative Stable-Isotope Probing (qSIP) with Metagenomics Links Microbial Physiology and Activity to Soil Moisture in Mediterranean-Climate Grassland Ecosystems. *MSystems*, *7*(6). <https://doi.org/10.1128/msystems.00417-22>
- Lee, S., Sieradzki, E. T., Nicolas, A. M., Walker, R. L., Firestone, M. K., Hazard, C., & Nicol, G. W. (2021). Methane-derived carbon flows into host-virus networks at different trophic levels in soil. *Proceedings of the National Academy of Sciences of the United States of America*, *118*(32), 1–8. <https://doi.org/10.1073/pnas.2105124118>
- Nicolas, A. M., Jaffe, A. L., Nuccio, E. E., Taga, M. E., Firestone, M. K., & Banfield, J. F. (2021). Soil Candidate Phyla Radiation Bacteria Encode Components of Aerobic Metabolism and Co-occur with Nanoarchaea in the Rare Biosphere of Rhizosphere Grassland Communities. *MSystems*, *6*(4). <https://doi.org/10.1128/msystems.01205-20>
- Pfeifer, E., Moura de Sousa, J. A., Touchon, M., & Rocha, E. P. C. (2021). Bacteria have numerous distinctive groups of phage–plasmids with conserved phage and variable plasmid gene repertoires. *Nucleic Acids Research*, *49*(5), 2655–2673. <https://doi.org/10.1093/nar/gkab064>
- Santos-Medellin, C., Zinke, L. A., ter Horst, A. M., Gelardi, D. L., Parikh, S. J., & Emerson, J. B. (2021). Viromes outperform total metagenomes in revealing the spatiotemporal patterns of agricultural soil viral communities. *ISME Journal*, *15*(7), 1956–1970. <https://doi.org/10.1038/s41396-021-00897-y>
- Starr, E. P., Shi, S., Blazewicz, S. J., Koch, B. J., Probst, A. J., Hungate, B. A., Pett-Ridge, J., Firestone, M. K., & Banfield, J. F. (2021). Stable-Isotope-Informed, Genome-Resolved Metagenomics Uncovers Potential Cross-Kingdom Interactions in Rhizosphere Soil. *MSphere*, *6*(5). <https://doi.org/10.1128/mSphere.00085-21>

Peer Review comments, second round -

Reviewer #2 (Remarks to the Author):

The authors have addressed all my comments, I think the manuscript could be accepted.

Reviewer #3 (Remarks to the Author):

With this revised manuscript, the authors have addressed all previous concerns with the original manuscript.

Response to Reviewers' Comments

Reviewer #2 (Remarks to the Author):

The authors have addressed all my comments, I think the manuscript could be accepted.

- *Thank you for your feedback.*

Reviewer #3 (Remarks to the Author):

With this revised manuscript, the authors have addressed all previous concerns with the original manuscript.

- *Thank you for your feedback.*